# AI-powered spatial cell phenomics enhances risk stratification in non-small cell lung cancer

Simon Schallenberg[1,23], Gabriel Dernbach[1,2,23], Sharon Ruane[2], Philipp Jurmeister [3], Cornelius Böhm [2], Kai Standvoss[2], Sandip Ghosh [2], Marco Frentsch[4,5], Mihnea P. Dragomir [1,6,7], Philipp G. Keyl [8], Corinna Friedrich [1,9], Il-Kang Na[4,5,6,10], Sabine Merkelbach-Bruse[11], Alexander Quaas[11], Nikolaj Frost [12], Kyrill Boschung[13], Winfried Randerath[13], Georg Schlachtenberger[14], Matthias Heldwein [14], Ulrich Keilholz[7,15], Khosro Hekmat[14], Jens-Carsten Rückert[16], Reinhard Büttner [11], Angela Vasaturo[17], David Horst [1], Lukas Ruff[2], Maximilian Alber[1,2], Klaus-Robert Müller [8,18,19,20] ✉ & Frederick Klauschen [1,3,8,21,22] ✉

Risk stratification remains a critical challenge in non-small cell lung cancer patients for optimal therapy selection. In this study, we develop an artificial intelligence-powered spatial cellomics approach that combines histology, multiplex immunofluorescence imaging and multimodal machine learning to characterize the complex cellular relationships of 43 cell phenotypes in the tumor microenvironment in a real-world retrospective cohort of 1168 non-small cell lung cancer patients from two large German cancer centers. The model identifies cell niches associated with survival and achieves a 14% and 47% improvement in risk stratification in the two main non-small cell lung cancer subtypes, lung adenocarcinoma and squamous cell carcinoma, respectively, combining niche patterns with conventional cancer staging. Our results show that complex immune cell niche patterns identify potentially undertreated high-risk patients qualifying for adjuvant therapy. Our approach highlights the potential of artificial intelligence powered multiplex imaging analyses to better understand the contribution of the tumor microenvironment to cancer progression and to improve risk stratification and treatment selection in non-small cell lung cancer.

In the last two decades, advanced lung cancer treatment has been revolutionized by targeted therapies against oncogenic kinases and immune checkpoints have significantly improved cancer-specific survival rates[1–4]. Despite this progress, lung cancer remains the leading cause of cancer-related deaths with an estimated 1.8 million fatalities worldwide in 2020[5].

Current initial patient stratification relies on the Union for International Cancer Control (UICC) TNM staging system. This system evaluates three key factors: tumor size (T); regional lymph node involvement (N), and distant metastasis (M). It is widely recognized for its objectivity and reproducibility and is the current gold standard in clinical practice[6,7].

While earlier UICC stages (I-III) are treated by surgery with curative intent, patients may receive adjuvant chemotherapy to improve survival because about half of these patients relapse and have an unfavorable prognosis. Between 21 and 71% of patients succumb within

5 years of diagnosis[7–12]. So far, the search for novel prognostic and predictive biomarkers to effectively stratify these early-stage patients for potential adjuvant treatment options has remained inconclusive, even with next-generation sequencing.

In contrast to the established staging and grading systems, as well as current mutational profiling focus on cancer cells, our study explores the impact of the tumor microenvironment (TME) on disease progression and its potential clinical value. The TME is a multifaceted ecosystem that includes a wide variety of immune and stromal cells embedded in a vascularized extracellular matrix and their spatial relationships among each other and with the cancer cells[13]. In recent years, the pivotal role of the TME in cancer initiation, progression, and therapeutic response has been recognized[14,15], and several studies have shown the prognostic significance of specific immune cell subsets[16–21]. In NSCLC specifically, the TME has emerged as a key prognostic factor. For instance, Backman et al. recently demonstrated that the individual densities of lymphocyte subsets, the combined abundance of distinct immune cell populations, and their distances to one another each contribute independently to prognostic information regarding clinical outcomes[22]. More recent studies further support this view by showing that the spatial distribution of regulatory and cytotoxic T cells is closely associated with tumor stage and disease progression in NSCLC[23,24]. However, most of these studies have mainly focused on selected cell types, cell counts, or cellular function within the TME, not taking into account their complex spatial relationships and heterogeneous distribution[15,25]. This, however, is critical for understanding the role of the TME in tumor pathology. In a landmark study, Galon et al.[26] for instance, showed that the localization of immune cells influences the outcome in colorectal cancer patients. This finding was subsequently validated in a large international multicenter study[27]. Similarly, Loi et al.[28] and Issa-Nummer et al.[29] associated a high degree of immune infiltration in the tumor stroma with an increased treatment response rate in breast cancer patients. The limitations of these studies are that they only analyzed a limited number of cell types or compared tumor and stroma regions at an aggregated level.

It is hypothesized that capturing and understanding the complexity of the TME requires a data-driven, integrated AI-based analysis. Exemplary, Keren et al.[30] explore this path on a small cohort by combining multiplex imaging with AI-driven image analysis and expanding the number of cell types analyzed at spatial resolution, allowing for a more comprehensive understanding of the composition and organization of the TME. While previous work has unraveled remarkable insights into the TME on smaller cohorts of well-selected, contrasting patients, these studies were limited in scope and patient diversity.

In this work, we cover the full routine spectrum of patients as they arrive in the clinic, allowing us to inspect the current tumor staging system and its interaction with the spatial cellular composition of the tumors. We show that complex spatial cellular relationships can be systematically analyzed to identify "cell niches" that provide clinically relevant information beyond the Union for International Cancer Control (UICC8) staging system and guide treatment decisions. Our study leverages a large bicentric NSCLC cohort, multiplex immunofluorescence-based (mIF) cell characterization, and AI-driven multimodal modeling. The approach combines classical histomorphology with multiplex immunofluorescence microscopy, including carcinoma and immune cell antigens, allowing for the classification of 43 distinct cell phenotypes. Our study comprises 1168 patients with surgically resected stage I-IV NSCLC for which we developed a scalable, AI-based automated analysis pipeline. The approach incorporates 14 distinct AI models for tissue segmentation, cell detection, and cell classification, followed by an explainable machine learning approach that combines cell phenotypes and cell localization to predict patient outcome. In total, we identify 53 million cells whose types and spatial locations are incorporated in the clinical AI model. Our analysis shows pronounced variations in cell density and composition between the major lung cancer subtypes adenocarcinomas (LUADs) and squamous cell carcinomas (LUSCs), as well as among patients within each subtype. Based on the cell density, we identify different carcinoma subtypes with specific immune states. For example, carcinomas with high cancer cell density and very low inflammatory activity, so-called "cold tumors", immunosuppressive carcinomas with or without inflammation, and carcinomas with B-cell-dominant inflammation. By combining cell phenotypes with cell localization, we identify 10 distinct spatially resolved cell neighborhoods, termed "cell niches", in LUADs and LUSCs. Using the cell niches, we finally train a predictor of patient-survival on the Berlin subcohort and validate it on the Cologne subcohort. Our study highlights the potential of combining a large clinical cohort, histology, multiplex immunofluorescence, and an integrated multimodal AI approach. This enables high-resolution spatial exploration of the TME and facilitates improved risk stratification for adjuvant therapy selection in lung cancer patients.

## Results

### Development of an AI-powered multimodal cellomics assay

For each of the 1168 NSCLC patients, four 1.5 mm tissue cores from representative tumor regions were selected and assembled to tissue microarrays (TMA) to enable high-throughput imaging. All patients had undergone surgical tumor resection between 2006 and 2019 in Berlin or Cologne with available information on histological tumor type, UICC8 stage, treatment, and clinical outcome (Supplementary Data 1).

A 12-plex immuno-fluorescence panel was used for TME characterization, including immune-related proteins (CD3, CD4, CD8, CD20, CD56, CD68, CD163, FOXP3, Granzyme B (GrB)), immune-checkpoints (PD-1, PD-L1), and cytokeratin (CK) as epithelial cell marker allowing for the distinction of 43 different cell types (see Methods section for details; Supplementary Data 2 and Supplementary Figs. 1–3). For an integrated analysis of the tumor histomorphology, the same sections were additionally stained with Hematoxylin & Eosin (H&E) and re-scanned. Subsequently, TMAs were split into individual spots and registered to single-cell precision, yielding a database of 57,000 tissue images (Fig. 1 and Supplementary Fig. 4).

While smaller-scale studies can be performed with off-the-shelf models, the scale of the data and its diversity in batch effects made custom model training necessary.

To this end, we trained a deep convolutional neural network with UNet architecture to segment carcinoma, necrosis, tumor stroma and healthy tissue[31]. Training was carried out iteratively in an active learning fashion with pathologists in the loop. The final model achieved a macro-averaged F1-score of 0.92 on tissue spots of hold-out cases. Recall and precision within each region are shown in Fig. 1B.

Cell detection was based on an optimized in-house version of StarDist[32]–a UNet-based object detection model using a star-convex polygon prediction layer to detect cell nuclei–that was fine-tuned on over 100,000 pathologist-annotated cell nuclei across several tissue types. The cell detection achieved an object-based test-set F1-score of 0.91 (Fig. 1C, Supplementary Fig. 5 and Supplementary Table 1). Finally, the cell classification task was modeled as a multi-label classification task with twelve independent models–one per mIF channel–using a ConvNext[33] architecture. We chose independent models per mIF channel to counteract "Clever Hans" effects[34], which we observed when using joint models that exploited marker correlations among the mIF channels. The cell classification models were applied to a total of 53 million cells and achieved an average F1-score of 0.91 on independent test cases (Fig. 1D). The resulting binary predictions were aggregated into a multi-hot vector per cell, which formed the basis for downstream niche classification. As outlined in Fig. 1E, niche composition was derived by comparing the local marker frequencies (within a 34 μm radius) to prototype distributions obtained via clustering. Full implementation details are provided in the Methods section.

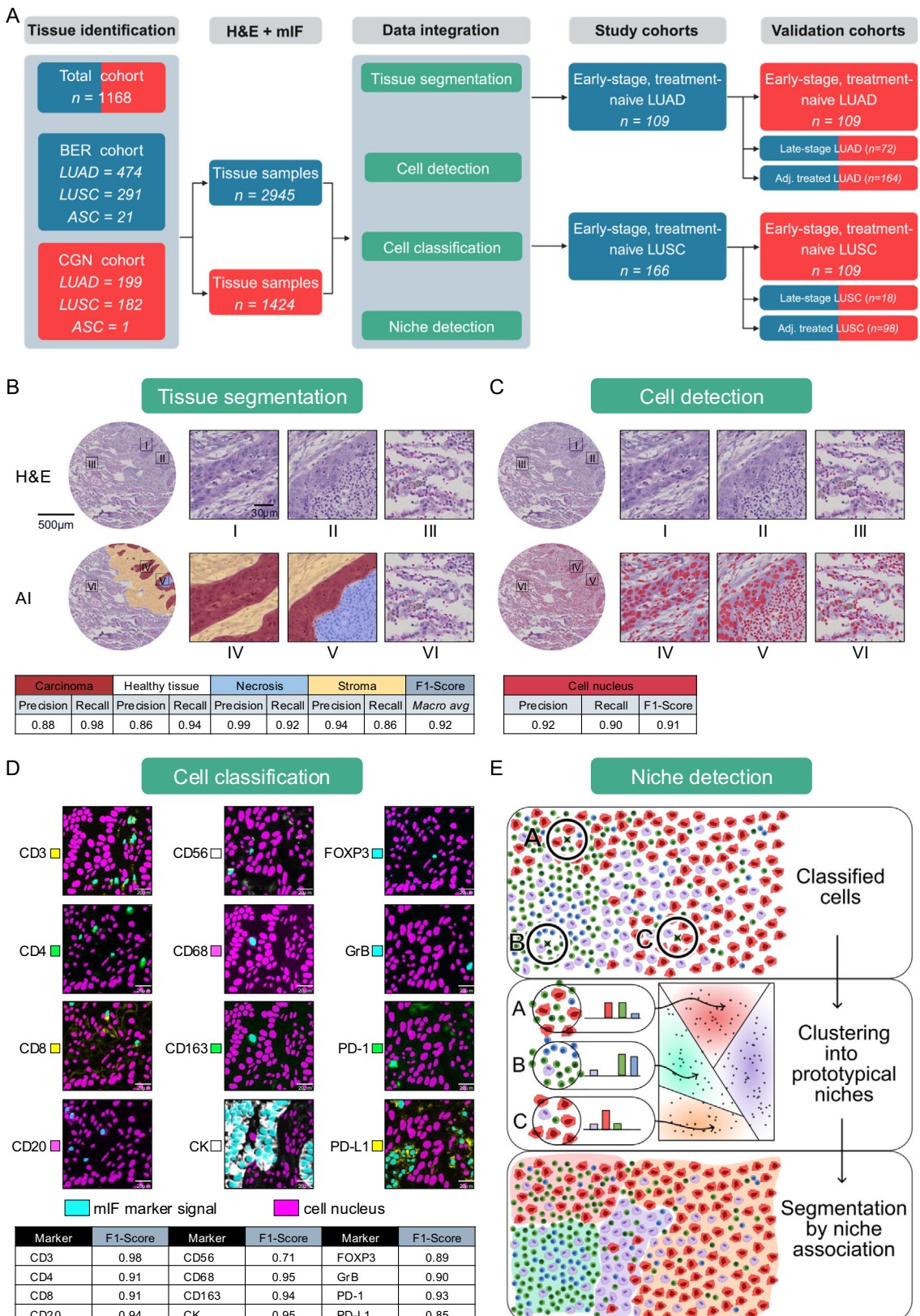

**Fig. 1 | Cohort description and cell phenomics analysis pipeline. A** Patient selection and data integration flow-chart. **B** Tissue segmentation model. Upper row: Original image (H&E), bottom row: AI-derived tissue segmentation with carcinoma in red, stroma in yellow, necrosis in blue and healthy tissue (uncolored). **C** Cell detection model. Upper row: Original image (H&E), bottom row: AI-derived cell detection (red polygons). **D** Twelve individual mIF-based cell classification models. **E** Niche detection: Upper box: For each cell, the number of cells within its 34 μm neighborhood was counted. Center box: All 53 million cell neighborhoods were clustered into ten distinct niches using neighborhood vectors separately for LUAD and LUSC. Lower box: The niche distributions were determined for every tumor. Scale bars: (**B** + **C**) 500 μm (tissue spots) and 30 μm (magnified regions); (**D**) 20 μm.

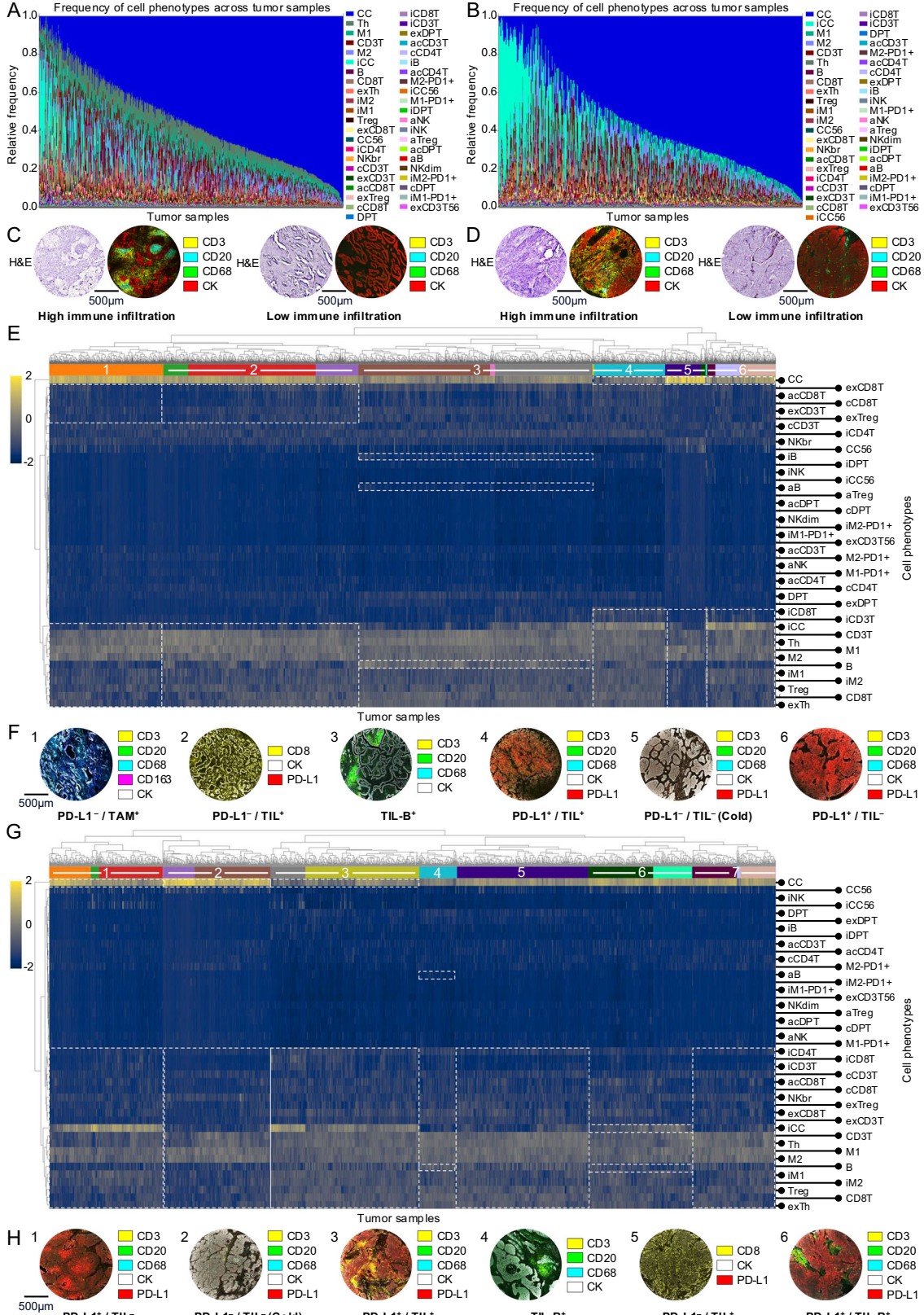

## Difference in immune composition of LUAD and LUSC on a cohort-level

We characterized the cellular landscape of NSCLC by quantifying the cellular composition based on the 43 mIF-derived cell types for both LUAD (Fig. 2A) and LUSC (Fig. 2B) separately (see Methods section for detailed explanation). Considerable variability in the quantitative composition of carcinoma and different immune cells could be observed in both carcinoma types (Fig. 2A, B). The spectrum ranges from immune cell-rich "hot" to immune cell-poor "cold" carcinomas (Fig. 2C, D). The observed diversity and extreme ends are in line with previous work on other cancer types, such as the work on triple-negative breast cancer from Keren et al.[30].

**Fig. 2 | Cell phenomics overview. A** Stacked bar plots of the abundances of cell phenotypes* for each tumor sample ($n = 2233$ from $n = 663$ patients) in LUAD, showing the highly variable distribution of immune cells and cancer cells across tumors. **B** Same as (**A**) but for LUSC ($n = 1659$ from $n = 462$ patients). In contrast to LUAD, a noticeable higher number of tumors exhibited high densities of immunosuppressive carcinoma cells (iCC). **C** Examples of tumors with high and low infiltration of immune cells and relatively low and high carcinoma cell content in LUAD. **D** Same as (**C**) but for LUSC. **E** Heatmaps sorted with hierarchical clustering in LUAD (z-scored and log-transformed for improved visibility; $n = 2233$ tumor samples from $n = 663$ patients. Horizontal axis: tumor samples; vertical axis: cell types. Arabic numbers highlight clusters corresponding to different clinically

relevant immune states (cf. **F**). Visual annotations highlight key patterns within the immune states. **F** Tumor examples with different clinically relevant immune states in LUAD. **G** Heatmaps sorted with hierarchical clustering in LUSC (z-scored and log-transformed for improved visibility; $n = 1659$ tumor samples from $n = 462$ patients. Horizontal axis: tumor samples; vertical axis: cell types. Arabic numbers highlight clusters corresponding to different clinically relevant immune states (cf. **H**). Visual annotations highlight key patterns within the immune states. **H** Tumor examples with different clinically relevant immune states in LUSC. Scale bars: (**C** + **D** + **F** + **H**) 500 μm. *Figure shows cell phenotype name abbreviations. For full names, see Supplementary Data 2. Source data are provided as a Source data file.

---

The impact of interactions between immune cells on cancer prognosis and response to checkpoint therapy is well recognized[35,36]. However, it remains a challenge to understand how specific (spatial) relationships among immune cell types impact clinical outcomes. This uncertainty is partly due to the limited accuracy of traditional manual scoring methods used in previous research. Our study addresses this issue with an AI-based analysis pipeline, improving the accuracy of our results.

To categorize LUAD and LUSC according to their unique immune states, hierarchical clustering was performed (Fig. 2E for LUAD and 2 G for LUSC). Examples of tumor samples with different immune states are shown in Fig. 2F (LUAD) and 2H (LUSC). Cellular compositions differed between LUAD and LUSC, reflecting their different histological types (Supplementary Table 2). In LUSC, carcinomas showed on average a higher density of carcinoma cells than in LUAD (1731.12/mm² vs. 1475.09/mm²; Chi-squared test, $p < 0.0001$****), which is consistent with the predominantly solid growth pattern of LUSC compared to the glandular growth of LUAD. Additionally, carcinoma groups with specific immune states were assembled that differed in their quantitative distribution between LUAD and LUSC (Chi-squared test, $p < 0.0001$****; for detailed cell phenotype density per cluster see Supplementary Data 3–6.). The classification of carcinomas by their specific immune status for prognosis and treatment response has been established by Mahmoud et al. 2011[37] and Taube et al. 2012[38]. Cluster group 6 in LUAD and cluster group 1 in LUSC were characterized by large proportions of immunosuppressive carcinoma cells (645.51/mm²; 878.65/mm²), alongside a lower frequency of T cells (307.71/mm²; 138.71/mm²) and B cells (9.16/mm²; 13.10/mm²), which is consistent with TME type 3 to tailoring cancer immunotherapeutic modules (PD-L1⁺/ TIL⁻), as defined in the literature[38–40]. TME type 3 represents a group of tumors for which PD-L1 positivity cannot be used as a predictive factor for response to Immuno-Oncology (IO) therapy. Instead, combination therapies including radiotherapy might be used to recruit lymphocytes into the tumor[40–44]. In contrast, cluster group 4 in LUAD and cluster group 3 in LUSC revealed TME type 1 (PD-L1⁺; TIL⁺) with a high frequency of both immunosuppressive carcinoma cells (726.65/mm²; 857.37/mm²) and lymphocytes (989.64/mm²; 1034.28/mm²). This tumor group shows response to IO therapy[11,33,34,36–38].

Cluster group 5 in LUAD and cluster group 2 in LUSC showed a high density of carcinoma cells (2340.06/mm²; 2116.97/mm²), with limited lymphocyte infiltrates (61.31/mm²; 94.07/mm²) and low frequency of immunosuppressive carcinoma cells (8.07/mm²; 4.85/mm²). This signature is consistent with the characterization of tumors with immunological ignorance (TME type 2) or "cold tumors", defined by a lack of T cells in the TME (PD-L1⁻; TIL⁻)[39,40,45]. The categorization of tumors into "hot" and "cold" types is increasingly recognized as a prognostic indicator for patient survival and as a predictor of response to immunotherapies[26,46–50]. Notably, LUSC had a higher proportion of "cold" tumors compared to LUAD (14.8% vs. 5.6%; Chi-squared test, $p < 0.0001$****).

Furthermore, the clustering identified a carcinoma group (LUAD cluster group 2; LUSC cluster 5) with higher lymphocyte ratios (435.34/

mm²; 585.89/mm²), alongside a low frequency of immunosuppressive carcinoma cells (10.57/mm²; 19.24/mm²), which is consistent with TME type 4 (PD-L1⁻/ TIL⁺)[39].

LUAD cluster group 3 and LUSC cluster 4 in particular were characterized by a high proportion of B cells (201.59/mm²; 156.74/ mm²). While immuno-oncology has primarily focused on T lymphocytes, there is growing evidence that tumor-infiltrating B cells (TIL-B) and plasma cells also play an important role in tumor biology, including NSCLC[51–58]. The higher frequency of B-cell-rich carcinoma subtypes in LUAD compared to LUSC (Chi-squared test, $p < 0.0001$****) aligns with current research findings[58,59].

Moreover, LUAD cluster 1 revealed low lymphocyte infiltration (182.56/mm²), accompanied with elevated levels of tumor infiltrating macrophages (TAM⁺) (184.51/mm²; monocyte-to-lymphocyte ratio = 1.01), whereas LUSC cluster 6 showed higher proportions of immunosuppressive carcinoma cells (71.51/mm²) and B cells (28.61/mm²), both of which are associated with patients' survival and treatment response[60,61]. We also identified a seventh cluster in LUSC that was less specifically defined, showing low levels of immunosuppressive carcinoma cells (9.17/mm²) and moderate B cell (19.73/mm²) and T cell infiltration (260.56/mm²).

The orthogonal validation of PD-L1 status via immunohistochemistry (IHC) and the lymphocyte count validation by H&E-based cell classification (Pearson correlation coefficient of $r = 0.8418/ r = 0.891$) underlines the accuracy of the approach (Supplementary Figs. 6, 7).

In summary, the results have shown that our multimodal AI-powered phenomics approach facilitates the automated, comprehensive, and accurate identification of prognostically relevant immune states within the TME at the cellular level.

## Composition and distribution of AI-derived spatial cell niches in LUAD

While the association of the non-spatially resolved cellular composition and clinical outcome is well established in the literature as described above, we hypothesized that analyzing the exact spatial relationships among cells in local cellular neighborhoods or "niches" may provide additional insights into biological behavior and clinical outcome.

To assess the spatial organization of the cellular landscape in LUAD, we computed the number of cells of the different cell types within a 34 micron neighborhood (Fig. 1E). The radius was selected to include both directly adjacent (first-order) and second-order neighbors in line with previous studies (see "Methods" for details)[30,62,63]. Using this approach, we identified 10 distinct cell neighborhoods, termed "cell niches". The composition of each niche was visualized using radar plots highlighting the proportions of the underlying expression markers (Fig. 3A–J).

Niche 1 was dominated by CK expressors with moderate levels of CD4⁺ and CD68⁺ cells (Fig. 3A). Niches 2 and 4 (Fig. 3B, D) exhibited a medium to high proportion of CK⁺ cells and an increased number of CD3⁺, CD4⁺, CD8⁺, and PD-1⁺ cells. In addition,

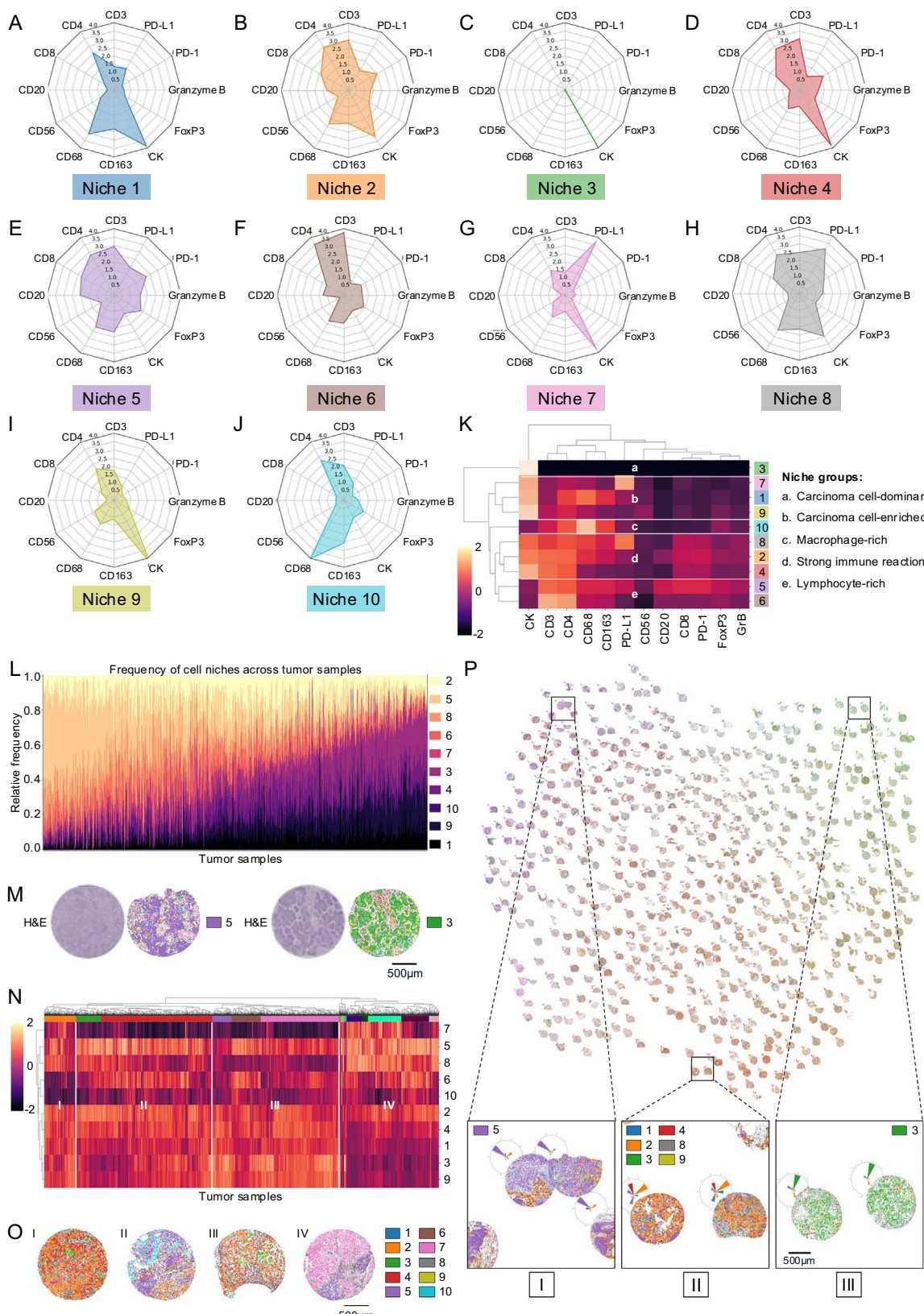

niche 2 had a higher proportion of CD68[+] cells compared to niche 4. Cells of niche 3 expressed almost exclusively CK (Fig. 3C), whereas niche 5 was characterized by a higher number of CD3[+], CD4[+], CD8[+], CD20[+], CD68[+], and FOXP3[+] cells as well as the expression of activation and exhaustion markers (PD-1[+], GrB[+]) (Fig. 3E). Niche 6 showed an increased proportion of CD3[+] and CD4[+] cells, along with

a low number of CK[+] cells (Fig. 3F). High proportions of cells within Niche 7 expressed CK and PD-L1 (Fig. 3G), while CK[+], PD-L1[+], CD3[+], CD4[+], CD8[+], CD68[+], and CD163[+] cells were moderately present in niche 8 (Fig. 3H). Niche 9 was characterized by high numbers of CK[+] cells in combination with low numbers of CD3[+], CD4[+], or CD68[+] cells (Fig. 3I). In contrast to all the other niches, niche 10 displayed a high

**Fig. 3 | Niche composition, distribution and patterns in LUAD. A**–**J** Visualization of niche compositions via radar plots highlighting the proportions of the underlying marker expressions. **K** Heatmap of cell niches sorted with hierarchical clustering for similarity in marker expression. Niches with similar marker expressions are labeled with lowercase letters. **L** Stacked bar plots of the abundances of cell niches across tumors in LUAD (n = 2233 tumor samples from n = 663 patients). **M** Tumor examples with enrichment of niche 5 (lymphocyte-rich, left) and niche 3 (carcinoma cell-dominant, right). Dots represent cells colored by their niche

membership. **N** Heatmaps sorted with hierarchical clustering for similarity in abundances, z-scored and log-transformed for improved visibility in LUAD. Tumor groups with differing niche patterns are labeled with roman numerals (n = 2233 tumor samples from n = 663 patients). **O** Examples of tumors with different niche patterns in LUAD as outlined in (**N**). **P** LUAD niche atlas for all tumor samples, see Supplementary Fig. 8 for a magnified version. UMAP projection of tumor samples according to niche patterns shows distinct groups (I, II, III; scale bar: 500 μm). Source data are provided as a Source data file.

proportion of CD68$^+$ cells, alongside a moderate number of CD163$^+$ cells (Fig. 3J).

To further compare the composition of the niches, we performed clustering based on the expressed markers (Fig. 3K). Firstly, we identified a group of three carcinoma cell-enriched niches with a low to moderate immune reaction (LUAD niches 1, 7, 9), characterized by a predominant presence of CK$^+$ cells alongside a low to moderate number of cells expressing lymphocyte-, macrophage-, and activation/exhaustion markers (e.g., CD3, CD20, CD68, PD-1, GrB). Secondly, another cluster comprising three niches revealed high CK expression, alongside high proportions of T cell markers (CD3, CD4, CD8), macrophage markers (CD68, CD163) and activation or exhaustion markers (PD-1, GrB), classified as carcinoma niches with strong immune reaction (LUAD niches 2, 4, 8). Furthermore, we found one carcinoma cell-dominant niche, in which cells expressed almost exclusively CK, clustered separately to each other niche (LUAD niche 3). Finally, three immune cell-dominant niches were identified, characterized by a high prevalence of immune cell markers and a low number of CK-positive cells (LUAD niches 5, 6, and 10). These were further categorized into two distinct lymphocyte-rich niches with predominant CD3 expression (LUAD niches 5 and 6) and one macrophage-rich niche with a high density of CD68$^+$ cells (LUAD niche 10).

The evaluation of niche type distribution across tumor samples showed considerable heterogeneity in LUAD (Fig. 3L). The tumors showed a broad spectrum of niche compositions ranging from a high prevalence of the lymphocyte-rich niche 5 (left side of the histogram; see exemplary H&E-stained tumor sample and niche overlay in Fig. 3M left) via mixed-niche type tumors to a high prevalence of the carcinoma cell-dominant niche 3 (right side of the histogram; see exemplary H&E-stained tumor sample and niche overlay in Fig. 3M right).

To evaluate the carcinomas in terms of their niche patterns, defined as the niche compositions within each tumor, we performed another hierarchical clustering analysis, this time over tumor samples and cell niches and identified four characteristic cluster groups (Fig. 3N). Cluster group I showed a mixture of carcinoma cell-dominant and carcinoma cell-enriched niches as well as niches with a strong immune reaction (niches 1–4, 8, and 9), whereas lymphocyte-rich and macrophage-rich niches were less frequent (exemplary tumor sample on the left in Fig. 3O). In contrast, cluster groups II and III exhibited a higher proportion of lymphocyte-rich niche 6 and macrophage-rich niche 10. Compared to cluster group III, cluster group II revealed a relatively low proportion of carcinoma cell-dominant or -enriched niches (niches 1, 3, and 9), with a higher proportion of niche 8, characterized by a strong immune reaction (exemplary tumor samples of cluster groups II and III in the middle left and right sections of Fig. 3O). Cluster group IV differed primarily in its higher proportion of carcinoma cell-enriched niche 7 and niche 8 (exemplary tumor sample on the right in Fig. 3O).

To obtain a comprehensive overview of the distribution and concentration of niche patterns, we visualized the tumor samples in a niche atlas—a two-dimensional UMAP projection in which the location of each tumor sample was determined by its niche pattern (Fig. 3P). Distinct groups of tumor samples with similar niche patterns became apparent that matched the groups as well as the clusters described above. For instance, groups of tumor samples showed enrichments of niche 5 (I), predominance of niche 3 (III) or were characterized by a

prevalence of niches 1–4, 8, and 9 (II), consistent with niche groups of the histogram (Fig. 3L) or cluster group 1 of the heatmap (Fig. 3N).

## Composition and distribution of AI-derived spatial cell niches in LUSC

Using the same approach as described above, we identified 10 different cell niches in LUSC. Niche 1 was characterized by high expression levels of CK along with moderate expression levels of CD3, CD4, CD68, and PD-L1, as well as activation and exhaustion markers PD-1 and GrB (Fig. 4A), while cells within niche 2 showed predominant CK expression, mixed with low proportions of CD68, CD3, and CD4 (Fig. 4B). Niche 3 had an intermediate number of cells, characterized by cells expressing CK, PD-L1, CD3, CD4, CD8, CD68, or CD163 (Fig. 4C). Niche 4 contained predominantly CK$^+$ carcinoma cells (Fig. 4D), while niche 5 showed an abundance of CD3$^+$ and CD4$^+$ cells (Fig. 4E). Within niche 6, most cells expressed CK and PD-L1 (Fig. 4F). Niche 7 showed a higher number of immune cell markers (e.g., CD3, CD20, CD68) as well as expression of activation and exhaustion markers (PD-1, GrB) (Fig. 4G). Niches 9 and 10 differed from the other niches in the high expression of macrophage markers (CD68, CD163) (Fig. 4I, J). Cells from niche 10 showed a mixture of CD68$^+$ and CD163$^+$ macrophages, whereas macrophages from niche 9 expressed predominantly CD163.

Similar to LUAD niche groups were found when comparing niche composition using cluster analysis (Fig. 4K). In contrast, we identified only two carcinoma-enriched niches with a low to moderate immune reaction (LUSC niches 2 and 6) and only two carcinoma cell niches with strong immune reaction (LUSC niches 1 and 3), characterized by a high expression of CK alongside low to high expression of immune cell, activation- or exhaustion markers, respectively. In addition, we again found one carcinoma cell-dominant niche (LUSC niche 4), consistent with LUAD niche 3. We further found five immune cell-dominant niches with a high proportion of immune cell markers and a low proportion of carcinoma cells (LUSC niches 5, 7, 8–10). These niches could be subdivided into three lymphocyte-dominant niches (LUSC niches 5, 7, and 10) and two macrophage-dominant niches (LUSC niches 8 and 9).

When analyzing the distribution of niches across tumor samples, we also found a pronounced heterogeneity in LUSC (Fig. 4L). In contrast to LUAD, fewer tumors showed only one dominant niche, but more combinations of two or three niches were observed (entropy within samples/clusters, scipy.stats.permutation_test, $p < 0.001$***/$p < 0.05$*). However, consistent with LUAD, a distinct group of carcinomas with a high prevalence of carcinoma-dominant niches (niches 2 and 4) was detected (left side of the histogram; see exemplary tumor sample in Fig. 4M left) as well as carcinomas with an enrichment of immune cells (niches 3 and 7) were found (right side of the histogram; see exemplary tumor sample in Fig. 4M right).

Similarly to LUAD, LUSC could be categorized into four cluster groups based on their niche patterns (Fig. 4N). Cluster group I was characterized by a higher proportion of the lymphocyte-dominant niches 5 and 7, macrophage-dominant niche 10, as well as niches with strong immune reaction (niches 1 and 3), and a lower occurrence of carcinoma cell-enriched and carcinoma cell-dominant niches 2, 4, and 6 (exemplary tumor sample on the left in Fig. 4O).

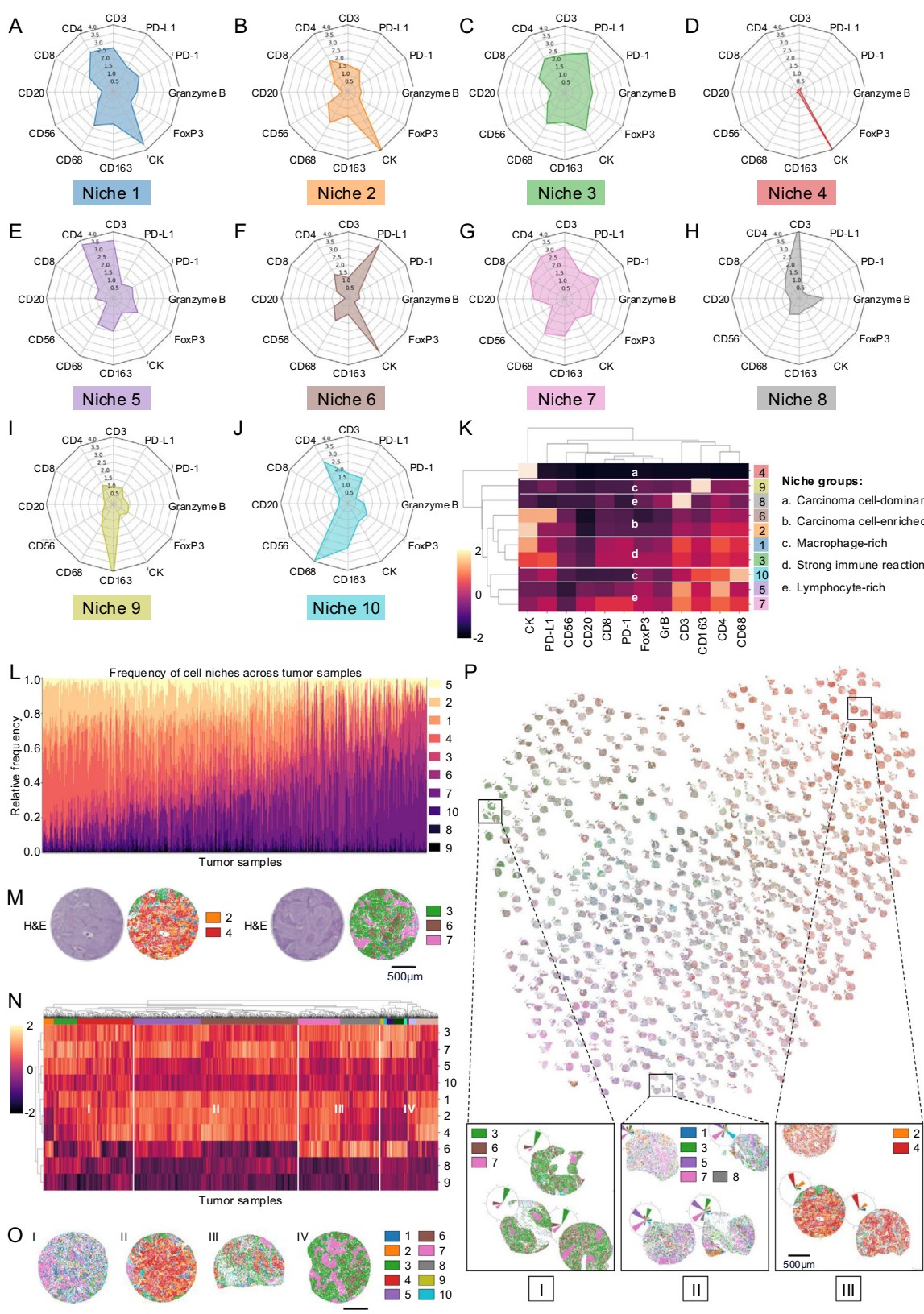

Cluster group II had a high proportion of niches 1–5 and 7 (exemplary tumor sample in the middle-left section of Fig. 4O). Cluster groups III and IV showed a higher prevalence of the carcinoma cell-enriched niche 6 compared to cluster groups I and II. In addition, both clusters had a higher proportion of niches 3 and 7, while niches 1, 2, and 4 were more prevalent in cluster III

(exemplary tumor samples of cluster groups III and IV in the middle-right and right sections in Fig. 4O).

The LUSC niche atlas (Fig. 4P) showed characteristic groups of tumors, reflecting the niche distributions and niche clusters described above. For example, groups were identified with a higher prevalence of niches 3, 6, and 7 (I), with a mixture of niches 2 and 4 (III), and with a

**Fig. 4 | Niche composition, distribution and patterns in LUSC. A–J** Visualization of niche compositions via radar plots highlighting the proportions of the underlying marker expressions. **K** Heatmap of cell niches sorted with hierarchical clustering for similarity in marker expression. Niches with similar marker expressions are labeled with lowercase letters. **L** Stacked bar plots of the abundances of cell niches across tumors in LUSC ($n = 1659$ tumor samples from $n = 462$ patients. **M** Tumor examples with enrichment of niches 2 and 4 (carcinoma cell-dominant and -enriched, left) and niches 3, 6, and 7 (immune cell-rich, right). Dots represent cells colored by their niche membership. **N** Heatmaps sorted with hierarchical clustering for similarity in abundances, z-scored and log-transformed for improved visibility in LUSC. Tumor groups with differing niche patterns are labeled with roman numerals ($n = 1659$ tumor samples from $n = 462$ patients. **O** Examples of tumors with different niche patterns in LUSC as outlined in (**N**). **P** LUSC niche atlas for all tumor samples, see Supplementary Fig. 9 for a magnified version. UMAP projection of tumor samples according to niche patterns shows distinct groups (I, II, III; scale bar: 500 μm). Source data are provided as a Source data file.

high proportion of niches 1, 3, 5, 7, and 8 (II) were identified, consistent with the niche groups from the histogram (Fig. 4L) and cluster group II from the heatmap (Fig. 4N).

In summary, these results show that we were able to comprehensively characterize the spatially resolved tumor composition at an unprecedented scale at single-cell level, revealing ten niches with unique cell neighborhood signatures in LUAD and LUSC, respectively, which could be further grouped according to their dominant immune status. Both entities exhibited carcinoma cell-enriched niches with a low to moderate immune reaction, carcinoma niches with strong immune reaction, carcinoma cell-dominant niches, as well as lymphocyte-dominant and macrophage-dominant niches. However, quantitative differences in the prevalence of these niche groups were critically observed between LUAD and LUSC.

## Translation of the niches into cell phenotype compositions for biological interpretability

Based on the distribution of the cell phenotypes (Supplementary Data 2) within each niche of LUAD and LUSC (Fig. 5A, B), we compared LUAD niches (Fig. 5C) with those of LUSC (Fig. 5D) based on their phenotype distribution (partial optimal transport; see Methods and Supplementary Fig. 10 for further details)[64,65]. In addition, we classified them according to their specific immune reaction and further grouped them into hot and cold niches (Fig. 5E).

LUAD niche 8 (Fig. 5C) and LUSC niche 3 (Fig. 5D) revealed higher proportions of immunosuppressive carcinoma cells (LUAD = 27%; LUSC = 31%) alongside a mixture of high immune cell infiltration (LUAD = 46%; LUSC = 47%), defining them as "inflamed niches" (Infl^high), and therefore as hot niches (Fig. 5E). LUAD niche 2 showed a balanced composition of carcinoma cells (51%) and the other half of immune cells (49%). The immune cells were a mixture of T lymphocytes (e.g., T helper cells (12%), CD8+ cells (3%), cytotoxic T cells (1%)) and macrophages (13%), corresponding to a hot niche with "strong immune infiltration" (Leuco^high). A similar phenotype distribution was observed in LUSC niche 1. LUAD niche 6 and LUSC niche 5 were also classified as hot niches. Both contained high densities of lymphocytes (LUAD = 88%; LUSC = 82%), while the number of carcinoma cells was low (LUAD = 1%; LUSC = 2%). We categorized them as "lymphoid rich I" (Lymph^high I). Both LUAD and LUSC had a unique hot niche. LUSC niche 8 was named "lymphoid rich II" (Lymph^high II) due to the high number of lymphocytes. The immune infiltrate of LUAD niche 4 consisted mainly of T lymphocytes (75%) and was therefore classified as "T cell dominant" (T cell^high). Both entities had a niche with an increased proportion of B cells (LUAD niche 5 = B cells 15%; LUSC niche 7 = B cells 10%). Furthermore, these niches were characterized by high proportions of T lymphocytes (LUAD niche 5 = 56%; LUSC niche 7 = 58%), regulatory T cells (LUAD niche 5 = 6%; LUSC niche 7 = 7%) and natural killer cells (LUAD niche 5 = 1%; LUSC niche 7 = 1%), moderate proportions of macrophages (LUAD niche 5 = 19%; LUSC niche 7 = 22%) and low levels of carcinoma cells (LUAD niche 5 = 3%; LUSC niche 7 = 2%). As shown in the example regions (Fig. 5C, D), both niches exhibited morphological features of tertiary lymphoid structures (TLS), consisting of organized cellular aggregates with T cell-rich zones and B-cell follicles[66]. Thus, both niches were classified as "tertiary lymphoid structures" (TLS^high).

We further identified five cold niches in each entity, four of which were found in both LUAD and LUSC, and one of which was found in

only one entity (Fig. 5E). We found an "immune deserted" niche (Immu^NULL) in LUAD (niche 3) and LUSC (niche 4), characterized by carcinoma cells without immune cell infiltrates. In addition, one "immunosuppressive excluded" niche (Immunosup^high) could be identified in LUAD (niche 7) and LUSC (niche 6), showing a high density of immunosuppressive carcinoma cells (LUAD = 65%; LUSC = 71%), alongside a low number of immune cells (LUAD = 6%; LUSC = 5%). Furthermore, we classified LUAD niche 9 and LUSC niche 2 as niches with weak immune infiltration (Leuco^low). Finally, we found a "TAM-dominated" niche (Mφ^high I) in each entity (LUAD niche 10; LUSC niche 10), which revealed a mixture of a high number of macrophages (LUAD = 92%; LUSC = 90%). In contrast to LUAD, LUSC had a second macrophage niche with a high proportion of M2 macrophages (93%), on the basis of which we classified it as "TAM-II-dominated" (Mφ^high II). LUAD niche 1 consisted of 81% carcinoma cells and 16% tumor-associated macrophages (TAMs) and was classified as "TAM-carcinoma" (LUAD Mφ^int). This niche was unique to LUAD.

In summary, we were able to contextualize the spatially resolved cell niches from a biological perspective. On the one hand, we found six distinct hot niches, of which two were characterized by heterogeneous lymphocyte populations (Lymph^high I, Lymph^high II), three showed inflammatory cell-rich carcinoma regions (Infl^high, Leuco^high, T cell^high) and one correlated in composition and morphology with TLS (TLS^high). On the other hand, we identified macrophage-rich neighborhoods (Mφ^high Ii, Mφ^high II), immunosuppressive (Immunosup^high)-, inflammatory cell-poor (Leuco^low, Mφ^int) as well as immune deserted (Immu^NULL) carcinoma regions, which could be summarized as cold niches. Eight of the twelve niches occurred in both LUAD and LUSC, while two niches were present only in one or the other carcinoma subtype.

## AI-derived cell-niches advance survival prediction beyond UICC8 standard

The UICC8 staging system, which categorizes tumors based on the surgical resection with respect to size, infiltration of histological structures and presence of lymph node and distant metastases, is a cornerstone of the standard of care and serves as a primary predictor of patient survival and a key determinant of treatment recommendations. However, UICC8-staging is limited, particularly with respect to predicting relapse in early-stage lung cancer and therefore cannot reliably stratify patients for adjuvant treatment selection. In our study, we found that the UICC8 system effectively stratifies cases of LUAD with a concordance index (c-index) of 0.633 and cases of LUSC with a c-index of 0.630, which is a measure of survival prediction accuracy. More precisely, the c-index[67] is the proportion of pairs of patients in which the patient with the higher risk value indeed experienced the event before the patient with the lower risk value. For prediction at random, the c-index is 0.5.

To investigate whether cellomics profiles improve patient stratification beyond the UICC8 baseline, we trained a survival predictor using cell densities within the tumor region and UICC8 data. To ensure predictive validity of the results, we developed the model on the Berlin patient cohort and validated it with the Cologne data set. Our models showed an improvement over UICC8 of 8% relative to a chance-level baseline of c-index = 0.5 for LUAD cases (c-index = 0.644, 95% CI [0.546−0.738], $p < 0.001$) and an improvement over UICC8 of 34% for LUSC cases (c-index = 0.674, 95% CI [0.565−0.773], $p < 0.001$). We

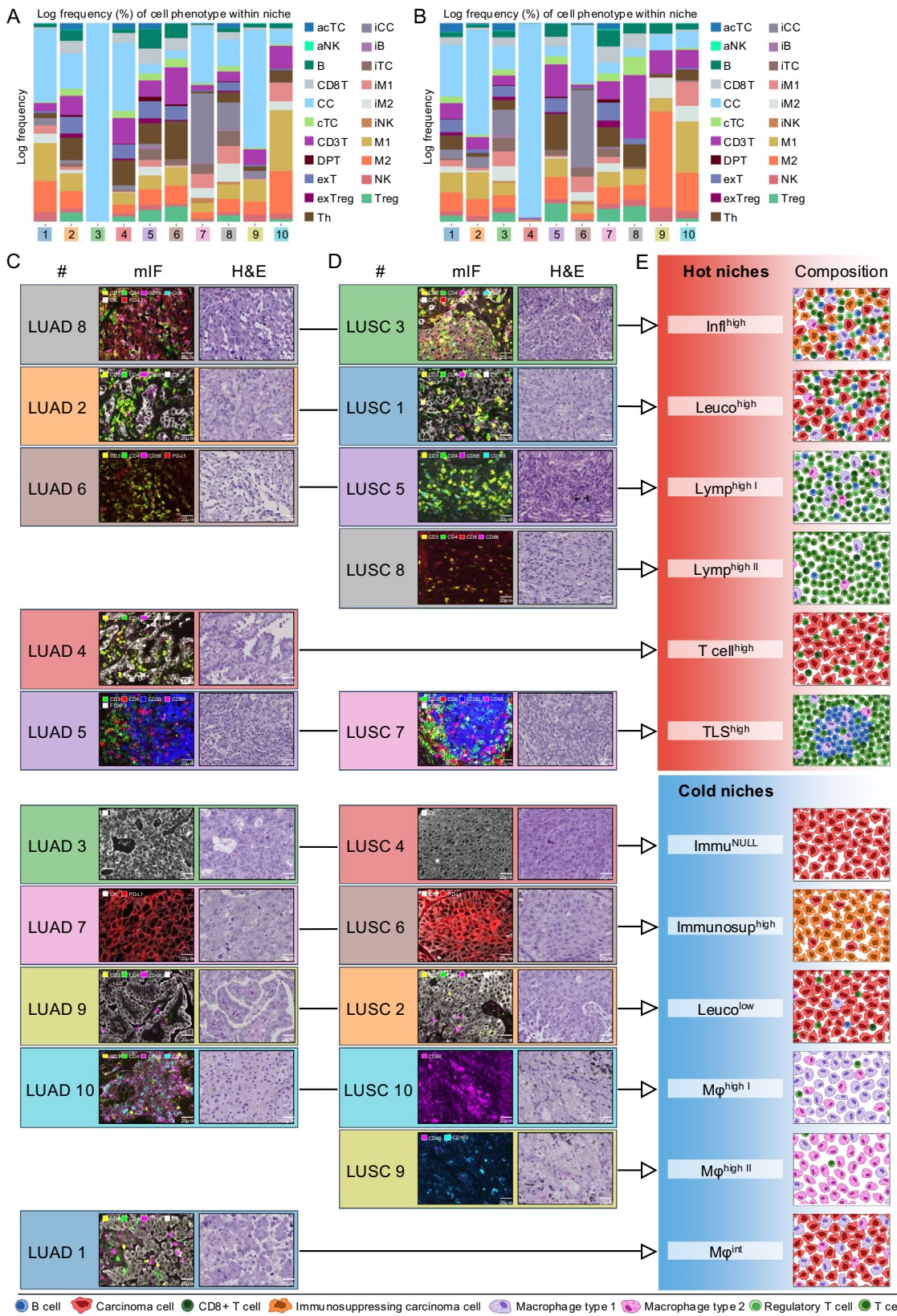

restricted our analysis to patients with complete tumor resection and without adjuvant treatment, as such treatments (radio- and chemotherapy) may influence the immune system and thus the patient's prognosis, and therefore distort the model results (for detailed patient selection flowchart see Supplementary Fig. 11). Integration of multiple TMA cores per patient was performed via max-pooling, i.e., the tumor

sample with the highest risk determines the overall risk of a patient ("hot-spot approach", see "Methods" section for further details). This followed the rationale that risk underestimation should be avoided. Of note, using average risk scores also improved patient stratification beyond UICC8 staging (Supplementary Table 3), but using maximum scores performed best.

**Fig. 5 | Niche composition and biological interpretation. A** Distribution of cell phenotypes within each niche in LUAD* (*n* = 2233 tumor samples from *n* = 663 patients). **B** Distribution of cell phenotypes within each niche in LUSC* (*n* = 1659 tumor samples from *n* = 462 patients). **C** Example images of LUAD niches in mIF and H&E stainings. **D** Example images of LUSC niches in mIF and H&E stainings. **E** Classification of cell niches with respect to their specific immune reaction, schematic representation of the composition in pictograms and grouping into hot and cold niches. The arrows connect identical niches in LUAD (**C**) and LUSC (**D**) with the corresponding classification (**E**), as determined by partial optimal transport analysis, which yields a unique solution and does not require repetitions (see Methods and Supplementary Fig. 10 for details). Scale bars: (**C** + **D**) 20 μm. *Figure shows cell phenotype name abbreviations. For full names see Supplementary Data S2. For detailed cell counts per niche in LUAD and LUSC, see Supplementary Tables 4 and 5, respectively. Source data are provided as a Source data file.

We performed the same analysis with the cell niches combined with the UICC8 as features and found a relative improvement over UICC8 of 14% in LUAD (c-index = 0.665, 95% CI [0.566–0.758]) and also a strong relative improvement of 47% in LUSC (c-index = 0.692, 95% CI [0.597–0.781]). These models that used cell niches significantly outperformed models just using cell densities (*p* < 0.01, Wilcoxon signed-rank test). A visualization of the model hold-out c-index bootstrap test distribution is presented in Figs. 6A and 7A. This finding supports the claim that the spatial organization of the TME is relevant for tumor progression and clinical outcome. This is also reflected in the survival analysis (see Kaplan–Meier plots in Fig. 6B for LUAD and Fig. 7B for LUSC), which confirms the superior predictive power of the niche-based models over models based on the UICC8 staging or cell densities alone. A visual comparison to the patient stratification by the UICC8 grading system can be found in Supplementary Figs. 12 and 13. Notably, patients assigned the lowest risk scores by our niche-base models have longer overall survival than patients with UICC8 stage 1. See Supplementary Table 3 for a comparison of the different models in LUAD and LUSC.

In summary, our findings suggest that spatial TME features have a significant prognostic impact that can be leveraged to improve UICC8-based clinical decision making.

### AI-based cell niche analysis identifies clinically relevant risk subgroups in stage 1 lung cancer

The improvement in survival prediction performance of the niche-based model comes with significant changes in patient risk assessment (Figs. 6C and 7C). In particular, a noticeable reclassification of the UICC8 stage 1 patient group could be observed—patients for whom surgery alone is the treatment of choice according to current guidelines. Approximately half of these patients were reclassified to a higher risk score based on niches and showed a prognosis similar to the group of patients originally classified as UICC stage 2—patients for whom adjuvant chemotherapy is recommended (Figs. 6D and 7D). Our study of the spatial composition of TME in early-stage LUAD and LUSC thus represents a paradigm shift in risk stratification by providing a significantly different scoring system that can effectively identify high-risk and potentially undertreated patients within a heterogeneous UICC stage 1 population.

As expected, the niche-based models developed on early-stage patients treated with surgery alone were able to stratify patients with late tumor stage or patients with adjuvant therapy with less precision. For patients with UICC8 = IV, the niche-based models stratified the LUSC subgroup into a high-risk class and a low-risk class (Supplementary Fig. 14). For patients with adjuvant therapy the niche-based models significantly separated LUAD into high and low-risks (Supplementary Fig. 15). Additionally, we compared risk groups with important clinical confounders[68] such as mutation status of frequently mutated genes in lung cancer as well as other known covariates, including sex, age, ECOG performance status, smoking status, tumor grade, thyroid transcription factor 1 (TTF-1) expression, and growth pattern, and found no relevant association (Supplementary Fig. 16. 17; Supplementary Data 7).

### Risk groups reveal distinct hot and cold niche patterns

To further understand the underlying biology of the proposed niche-based stratification, we analyzed the cell niche patterns associated with the three risk groups in LUAD (Fig. 6E–G) and LUSC (Fig. 7E–G).

LUAD with niche risk score 1 (RS1) showed a predominant mixture of hot niches (Leuco^high, Infl^high, T cell^high), which is consistent with the literature. Several studies have shown that high densities of tumor-infiltrating lymphocytes (TILs)—such as CD3+TILs, CD4+TILs, CD8+TILs and CD20+TILs – are favorable prognostic biomarkers[17,22,69–71]. In line with the LUAD-specific analysis by Sorin et al.[72], two B–cell–rich niches with high regulatory T cell content (TLS^high, Lymp^high I) were also enriched in our high-risk groups RS2 and RS3, further supporting their association with poor prognosis in LUAD. In addition, a higher concentration of the cold TAM-dominated niche (LUAD Mφ^high) was observed at the higher risk scores. This is consistent with the observation that an increased incidence of TAMs is associated with reduced overall survival in various cancer types, and aligns with reports by Backman et al.[22] and Desharnais et al. (LUAD)[73] linking macrophage-rich niches to poor prognosis[60,74]. We also identified additional cold immune niches (Immu^NULL, Immunosup^high, and Leuco^low) characterized by high densities of tumor or immunosuppressive tumor cells, which were strongly enriched in RS3 and associated with particularly poor prognosis—patterns not reported in the previously cited large-scale spatial studies.

Similar to LUAD, for LUSC, higher concentrations of hot niches (TLS^high, LUSC Infl^high) were found at risk score 1, whereas the cold niches Leuco^low and Immu^NULL accumulated at risk scores 2 and 3. Again, cold macrophage niches (Mφ^high I; Mφ^high II) were enriched in high-risk tumors. In contrast to LUAD, TLS^high niche and Immunosup^high niche were both particularly observed in risk score 1 tumors, whereas Chen et al.[75] reported a favorable association of TLS enrichment in NSCLC without histology-specific separation. This underlined the distinct nature of LUAD and LUSC. We also observed that the B-cell- and regulatory T cell-rich niche Lymp^high I was enriched in RS1, whereas a second such niche (Lymp^high II) was enriched in higher-risk groups, emphasizing entity-specific prognostic associations and highlighting the importance of considering niche patterns rather than single niches for prognostic prediction.

In a complementary analysis, we quantified fibroblast abundance from H&E images and analysed their distribution across the cell niches. Fibroblast abundance showed weak to moderate correlations with niche frequencies and was significantly higher in "hot" niches compared to "cold" niches in both LUAD and LUSC, but did not show a statistically significant association with patient survival (Supplementary Figs. 18 and 19).

In summary, the results highlight that AI-derived analysis showed distinct niche patterns associated with different patient survival risk scores in both LUAD and LUSC. Both LUAD and LUSC showed an accumulation of hot niches at risk score 1, whereas cold niches were found in higher concentrations at risk score 2 and 3, consistent with findings from the literature.

### Risk-score heterogeneity and validation on whole tumor sections

To evaluate the adequacy of TMAs for risk score assessment, we evaluated and compared the intra- and inter-tumoral heterogeneity of the niche-based risk scores for the different tissue cores for each patient and across the whole cohort. The results demonstrated a low intra-tumoral vs. inter-tumoral risk score variation (Supplementary Fig. 20). To further evaluate the use of TMA cores as surrogates for whole tumor sections, we compared the risks derived from 20 NSCLC

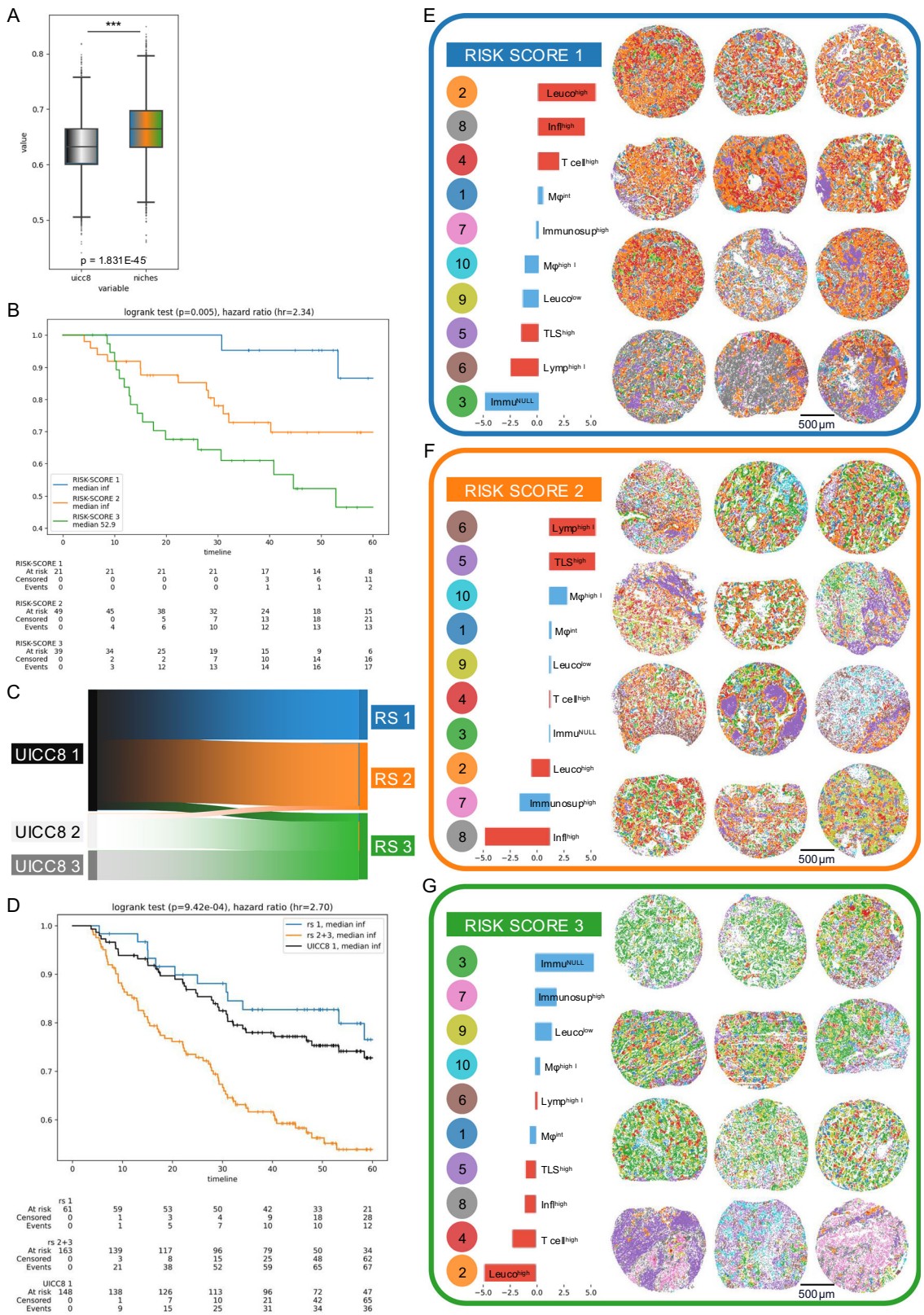

surgical whole tissue sections with the risk assessment based on their corresponding TMA cores. The results demonstrated high correlation of risk scores and 100% concordance on the level of risk groups (Fig. 8A–D; Pearson's $r = 0.979$; $p < 0.0001$****) and LUSC (Fig. 8E–H; Pearson's $r = 0.997$; $p < 0.0001$****). A complete visualization of the whole tumor sections can be found in Supplementary Figs. 21–23.

## Discussion

Understanding the complexity of the TME and its relevance for tumor biology and therapeutic strategies remains a major challenge in cancer research and oncology.

In the present study, we built on previous work highlighting the prognostic value of the TME, such as the studies by Chen B. et al.[17] and Kinoshita T et al.[18] and recent evidence suggesting that the spatial

**Fig. 6 | Survival prediction and risk scores in LUAD.** Results for Cologne patients for model trained on Berlin cohort. **A** Bootstrap distribution of c-index comparing UICC8 staging versus niche enhanced risk-scores ($n = 109$ patients; $p < 0.001$, two-sided Mann–Whitney U test). Box plots show the median (center line), interquartile range Q1–Q3 (box), and whiskers to the most extreme points within 1.5× IQR; points beyond are shown as fliers. **B** Kaplan–Meier analysis, patients ($n = 109$ (same as (6 A)) stratified by the risk scores as assigned via the cell-niche survival model. Patients are split into tertiles for better comparison to UICC8 (see Supplementary Fig. 12 for a corresponding Kaplan-Meier analysis based on the UICC8 model). Separation tested significantly via log-rank test. **C** Sankey plot showing the redistribution of patients ($n = 109$) from the UICC8 to the niche-based risk stratification. A large number of the UICC8 stage 1 patients was reclassified as risk score 2 (RS2) indicating potentially undertreated high-risk patients. **D** Kaplan–Meier analysis for UICC8 stage 1 patients ($n = 148$; black curve) and the restratification by risk score (RS1 = blue curve, $n = 61$ patients; RS2 = orange curve, $n = 163$ patients). Separation tested significantly via log-rank test. **E** Example tumors ($n = 12$) with the characteristic niche patterns of risk score 1. Colored dots with niche numbers in legend (left) correspond to dots in tumor samples representing cells colored by their niche membership (right). Bar plots indicate relative abundance of niches in tumors: Hot niches are indicated by red, cold niches by blue bars. Niches are sorted by the abundance within the risk-score group relative to the global average, e.g., the Leuco$^{high}$ niche was observed more often than in the whole cohort whereas Immu$^{Null}$ was observed less frequently. **F** Same as (**E**) but for risk score 2. **G** Same as (**E**) but for risk score 3. Scale bars: (**E**–**G**) 500 μm. Source data are provided as a Source data file.

arrangement of cells within tumors critically influences disease progression and therapeutic response[22,30,76]. To this end, we developed a multimodal AI-powered TME profiling approach that revealed spatial cellular "niche" patterns predictive of biological tumor behavior in lung cancer.

The strength of our approach lies in its ability to identify clinically relevant spatial cellular neighborhoods by combining multiplex imaging with multimodal AI. This method extends traditional tumor categorizations by considering the microscale interactions between different cell types in the TME. Our AI-driven models predict patient outcomes with higher accuracy than the current clinical gold standard (the UICC8 staging system), suggesting a potential paradigm shift in the stratification of NSCLC. This could lead to more personalized treatment strategies where therapies are tailored to the specific cellular composition of individual tumors, representing a significant advance towards precision medicine in oncology.

Our findings support the hypothesis that not only the quantitative composition of the immune and stromal cells matter, but that also their spatial organization within the TME is critical to understanding tumor behavior. Incorporating these spatial patterns into the diagnostic process facilitates the identification of low- and high-risk subgroups in lung cancer patients that are currently classified as low-risk by UICC8 staging alone. This is clinically highly relevant because stage 1 lung cancer patients do not receive adjuvant therapy according to current guidelines. However, the stage 1 high-risk subgroup we identify has a similar risk profile as stage 2 patients and would therefore require adjuvant therapy.

Although high-quality mIF imaging would currently only be available at specialized centers, the fact that our TME profiling relies on a robust 12-plex mIF approach makes its broader diagnostic implementation realistic for the near future.

Moreover, while we trained and validated the approach on two independent lung cancer patient cohorts from two separate centers in Germany, further evaluations will be needed to promote clinical implementation. Also, it remains to be shown if the observed niche patterns represent general microenvironmental properties and could be transferred to other cancer types.

Future research should aim to validate and refine the predictive power of cell niche patterns in broader patient populations. In this context, cancer-associated fibroblasts (CAFs) represent a particularly relevant stromal cell population, as recent work by Cords et al.[76] demonstrated that distinct CAF phenotypes are linked to either favorable or poor prognosis in NSCLC. While our current analysis did not show any significant association between the abundance/density of fibroblasts in cell niches with risk scores, we were not able to take into account different CAF subtypes because fibroblast subtype markers were not included in our mIF panel, which we designed to limit complexity for easier clinical applicability. Future studies will have to evaluate the additional clinical value of incorporating such phenotypic distinctions. Future work should also integrate multi-omics data to gain a deeper insight into the mechanisms underlying niche formation and its relevance on tumor progression. Model systems allowing experimental perturbation of the identified cell niches could establish causal relationships between niche patterns, tumor behavior and response to treatment, thus revealing potential therapeutic targets.

In summary, by identifying cell niches and demonstrating their clinical value, our study not only contributes to the understanding of the role of the TME in NSCLC progression but also exemplifies the transformative potential of AI-powered multiplex technologies. This study paves the way for sophisticated diagnostic and therapeutic approaches that take into account complex spatial tumor properties, potentially leading to more effective and personalized treatment strategies.

## Methods
### Patient population and dataset
We examined 1168 NSCLC patients treated from 2006 to 2019 at the University Hospital Cologne ($n = 382$) and Charité Berlin ($n = 786$), comprising 730 males and 438 females, with a mean age of 66.1 years and a median age of 66.4 years. 346 patients had received adjuvant treatment. We had definite survival times for all but 209 patients that were still alive at the time of analysis and covered observation periods of a maximum of 186 months. We divided NSCLC patients into 673 LUAD, 473 LUSC, and 22 excluded cases with mixed subtype. Diagnoses, histological growth patterns, tumor grading, pTNM classification, angioinvasion, lymphatic invasion, and tumor staging (according to the 8th edition of the TNM classification, AJCC) were reviewed by two board-certified pathologists with over 6 (S.S.) and 10 (F.K.) years of experience in thoracic pathology. Histological subtyping was based on hematoxylin and eosin (H&E), Periodic Acid–Schiff (PAS), and Van Gieson staining. Routine diagnostic immunohistochemical panels used for tumor classification included p40, p63, CK5/6, CK7, and TTF-1 (Supplementary Fig. 24). The diagnosis of LUAD was based on the presence of glandular growth patterns and typically showed CK7 positivity and p63/p40 as well as CK5/6 negativity, with or without co-expression of TTF-1. LUSC was diagnosed in the presence of keratinization and/or intercellular bridges in combination with positivity for p40, p63, and CK5/6 and no or only weak CK7 expression. All major histological subtypes[77] of invasive non-mucinous adenocarcinoma—namely lepidic, acinar, papillary, micropapillary, and solid patterns—as well as invasive mucinous adenocarcinomas were included in the study (see Supplementary Fig. 24). Rare LUAD variants[77], including minimally invasive adenocarcinoma, colloid adenocarcinoma, fetal adenocarcinoma, and enteric-type adenocarcinoma, as well as large cell lung carcinomas were excluded due to low frequency and distinct biological characteristics. Neuroendocrine tumors—identified by positivity for synaptophysin and chromogranin A or CD56—including small cell lung carcinoma (SCLC), large cell neuroendocrine carcinoma, and typical or atypical carcinoids, were excluded from all analyses because they represent a separate tumor group as defined by the WHO.

### Rationale for separate analysis of LUAD and LUSC
We performed separate analyses for LUAD and LUSC based on clinicopathological and methodological rationales.

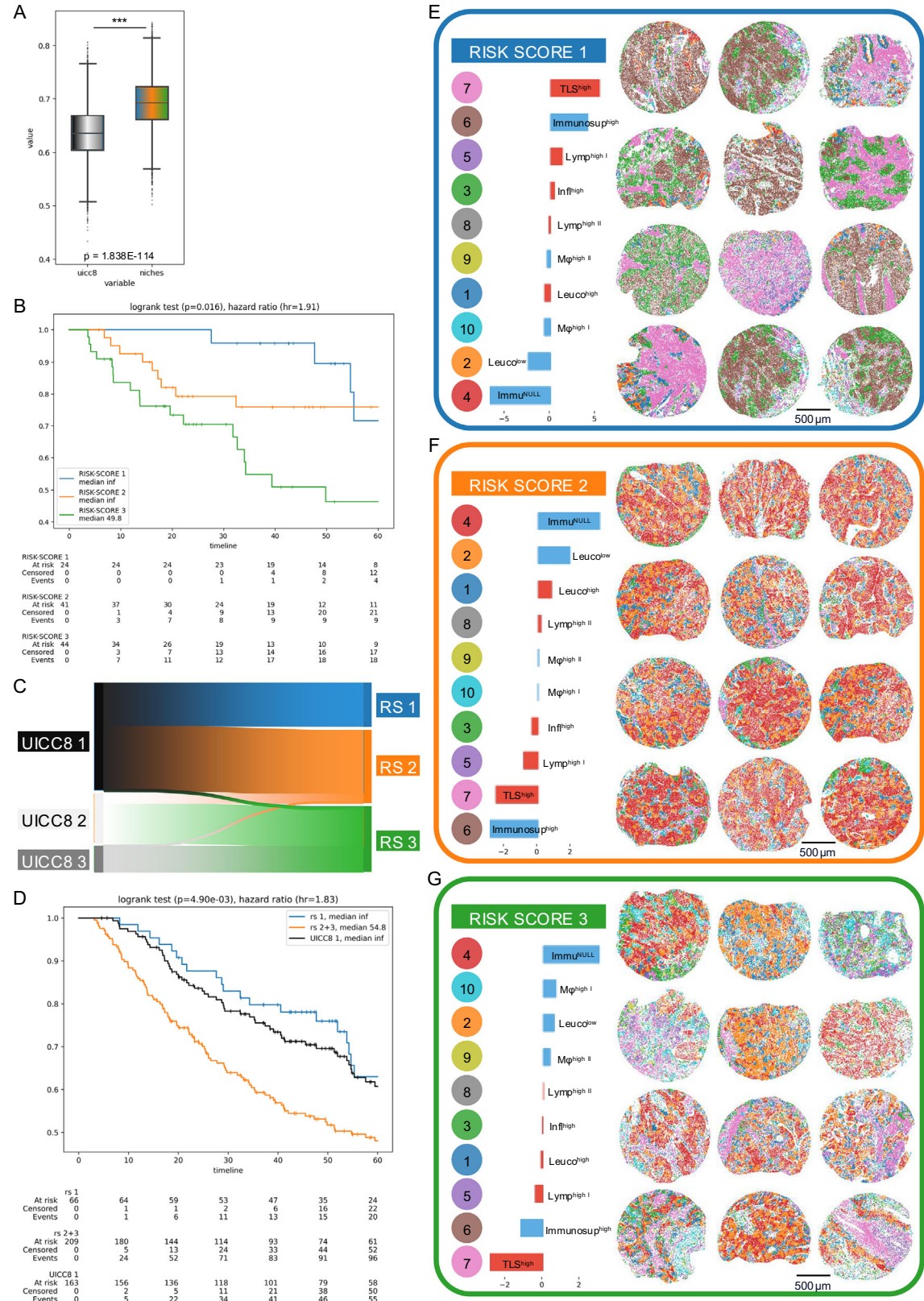

## Clinicopathological rationale

LUAD and LUSC are increasingly recognized as distinct tumor entities, based on well-established differences in etiology, histology, and molecular biology:

1. Different etiology: In contrast to other types of lung cancer, LUAD also manifests in non-smokers, particularly in female non-smokers. In contrast, LUSC is strongly associated with smoking[78].

2. Different origin: LUSC typically arises from a main or lobar bronchus, with approximately two-thirds of cases occurring in the central lung compartment. In contrast, LUAD is more frequently observed in peripheral regions[77].

3. Different pathogenesis: The atypical adenomatous hyperplasia and adenocarcinoma in situ, which originate from type II pneumocytes and/or club cells, are regarded as the precursors of

**Fig. 7 | Survival prediction and risk scores in LUSC.** Results for Cologne patients for model trained on Berlin cohort. **A** Bootstrap distribution of c-index comparing UICC8 staging versus niche enhanced risk-scores ($n = 109$ patients; $p < 0.001$, two-sided Mann–Whitney U test). Box plots show the median (center line), interquartile range Q1–Q3 (box), and whiskers to the most extreme points within 1.5× IQR; points beyond are shown as fliers. **B** Kaplan–Meier analysis, patients ($n = 109$) stratified by the risk scores as assigned via the cell-niche survival model. Patients are split into tertiles for better comparison to UICC8 (see Supplementary Fig. 13 for a corresponding Kaplan–Meier analysis based on the UICC8 model). Separation tested significantly via log-rank test. **C** Sankey plot showing the redistribution of patients ($n = 109$) from the UICC8 to the niche-based risk stratification. A large number of the UICC8 stage 1 patients was reclassified as risk score 2 (RS2) indicating

potentially undertreated high-risk patients. **D** Kaplan–Meier analysis for UICC8 stage 1 patients ($n = 163$; black curve) and the restratification by risk score (RS1 = blue curve, $n = 66$ patients; RS2 = orange curve, $n = 209$ patients). Separation tested significantly via log-rank test. **E** Example tumors ($n = 12$) with the characteristic niche patterns of risk score 1. Colored dots with niche numbers in legend (left) correspond to dots in tumor samples representing cells colored by their niche membership (right). Bar plots indicate relative abundance of niches in tumors: Hot niches are indicated by red, cold niches by blue bars. Niches are sorted by the abundance within the risk-score group relative to the global average, e.g., TLS$^{High}$ the niche was observed more often than in the whole cohort whereas Immu$^{Null}$ was observed less frequently. **F** Same as **E** but for risk score 2. **G** Same as (**E**) but for risk score 3. Scale bars: (**E**–**G**) 500 µm. Source data are provided as a Source data file.

invasive LUAD[79,80]. In contrast, squamous dysplasia and squamous carcinoma in situ are considered to be the precursors of LUSC, which arise in the bronchial epithelium[77].

4. Different histology: The histomorphology of LUAD and LUSC is markedly disparate, exhibiting distinct cytomorphology, growth patterns, and immunohistochemical expression profiles[81–83].

5. Different molecular landscape: LUAD and LUSC exhibit markedly disparate molecular signatures[84,85]. For instance, several oncogenic driver gene alterations, such as EGFR or ALK have been identified in LUAD, which are typically not present in LUSC.

6. Different treatment effects: Some targeted agents developed for LUAD have been shown to be largely ineffective against LUSC. In addition, studies have shown that the survival benefit of IO therapy is histology-dependent[86].

## Methodological rationale

Joint niche discovery would have required assuming shared spatial biology between LUAD and LUSC, potentially obscuring subtype-specific patterns. To preserve biological resolution, we performed niche discovery separately. For comparability, we aligned niches post hoc at the phenotype level using partial optimal transport to identify matched niche pairs across subtypes (see also "Methods", p. 37).

## Tissue microarray construction and staining

For each patient, 4 cores of 1.5 mm diameter representative tumor tissue were taken and placed on TMA slides, each containing a tonsil stain as a positive control for staining success and spot to patient mapping verification. TMAs were stained with an antibody panel for use in multiplex immunofluorescence imaging by Ultivue (Cambridge, MA).

The 12-plex immunofluorescence panel was designed to capture multiple immune cell phenotypes present in the NSCLC TME (pan-cytokeratin [panCK], CD3, CD8, CD68, CD20, CD4, FoxP3, PD-1, PD-L1, CD56, CD163, GranzymeB). Multiplex immunofluorescence (mIF) staining was performed on 2 µm sections of formalin-fixed paraffin-embedded (FFPE) tissue blocks of NSCLC TMA on the Leica Bond RX autostainer as previously described in Vasaturo et al.[87]. In brief, FFPE samples are dewaxed, and epitope retrieval is performed. After blocking, the slides are incubated with DNA-barcoded primary antibodies. All antibodies were provided in pre-optimized Ultivue kits and were diluted 1:100 in antibody diluent according to the manufacturer's standard protocol. All the targets are simultaneously amplified to increase assay sensitivity, and then, for each barcode, a complementary oligonucleotide probe tagged with a fluorescent dye is used to label the antibody conjugates. Different fluorophores are associated with each barcode–probe pair to spectrally separate the targets during imaging. High-plex panels require an Ultivue proprietary process, termed "exchange", which gently removes the fluorescent signal by dehybridizing the first set of four probes, while leaving the conjugated antibodies intact and bound to the tissue for detection by a second set of probes. A 12-plex assay thus consists of three image detection rounds using four probes per round, generating 4 × 3 images

of the same tissue using the AxioScan Z1 whole slide scanner. Since tissue spots can change position during the different scanning phases, we collected per-spot ROI annotations of each TMA to ensure that each spot is assigned to the right patient information. Splitting the TMAs into individual spot images greatly facilitated later image registration efforts. Registration was based on a DAPI signal that was contained in each scan group.

In total, the resulting dataset comprised a total of 57,000 microscopy images with an average size of 8650 × 8350 for the mIF channels and 17300 × 16700 for the H&E slide, representing a region of around 1.76 square millimeters each.

| Antibodies | Company | Identifier | Dilution |
|---|---|---|---|
| Anti-human CD3 | BioCare | ACI 3170 A | 1:100 |
| Anti-human CD4 | Abcam | ab238798 | 1:100 |
| Anti-human CD8 | BioCare | ACI3160CF | 1:100 |
| Anti-human CD20 | Thermo/eBio | 14-0202-82 | 1:100 |
| Anti-human CD56 | Thermo/eBio | 701379 | 1:100 |
| Anti-human CD68 | BioCare | CM033CF | 1:100 |
| Anti-human CD163 | Abcam | ab213612 | 1:100 |
| Anti-human FoxP3 | Thermo/eBio | 14-4777-82 | 1:100 |
| Anti-human PanCK | Bethyl/Fortis | PGS170523 | 1:100 |
| Anti-human PD-1 | Abcam | ab251613 | 1:100 |
| Anit-human PD-L1 | Abcam | ab226766 | 1:100 |
| Granzyme B | Abcam | ab214443 | 1:100 |

## Contact for reagent and resource sharing

Additional information and requests for reagents should be directed to Ultivue (https://ultivue.com).

## Imaging quality control

A pathologist reviewed each of the 4400 H&E scans to ensure that spots with tissue damages and tears, that potentially occur during staining, are rejected from further analysis, affecting around 10% of data. For each mIF-channel, the intensity histograms of all spots were collected and analyzed with a local outlier factor analysis as implemented in scikit-learn to highlight potential glow-artifacts and auto-fluorescence. The top 1% highest scoring tissue spots were inspected for rejection bty a pathologist, affecting less than 0.1% of data that was dropped from further analysis.

## Registration and tissue alignment quantification

For every tissue spot, each mIF scan group of four channels was registered to the H&E baseline using the DAPI nuclear expression signal as a guide. We employed a proprietary registration algorithm developed by Aignostics and applied it at full gigapixel resolution to

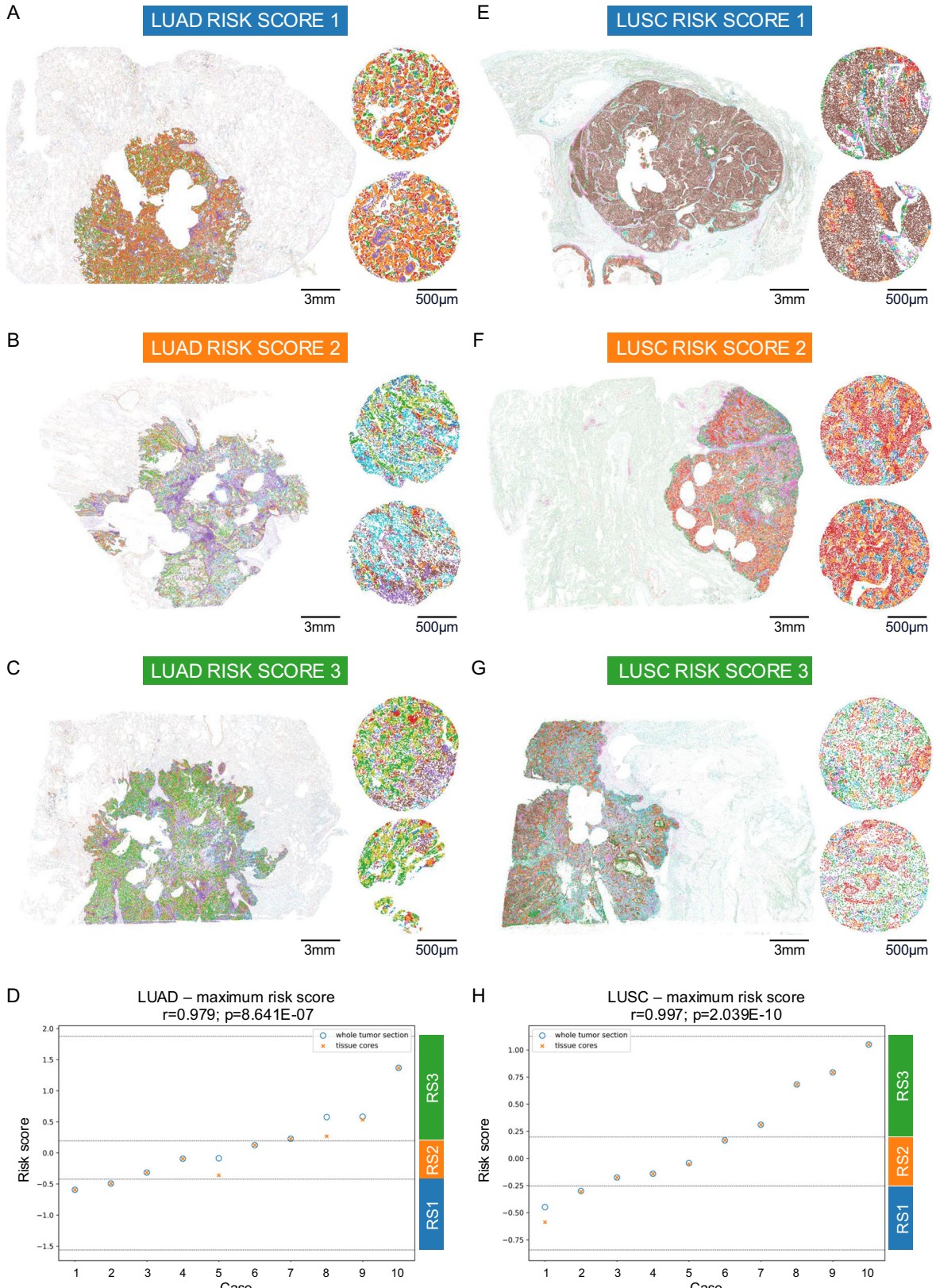

**Fig. 8 | Niche patterns on whole tumor sections. A** Whole LUAD section (left) and corresponding tumor cores (right) with niche patterns of risk score 1. **B** Same as (**A**) but for risk score 2. **C** Same as (**A**) but for risk score 3. **D** LUAD risk score concordance (*n* = 10 patients): Risk scores (maximum risk) for whole tumor sections (blue circles) are highly correlated with tissue core risk scores (orange crosses). Risk group assignments (RS1-3) are concordant with *r* > 0.97 (two-sided Pearson's correlation, scipy.stats). *y*-axis: risk score, *x*-axis: cases (ordered by increasing final

risk score), horizontal lines show boundaries between risk scores. **E** Whole LUSC section (left) and corresponding tumor cores (right) with niche patterns of risk score 1. **F** Same as (**D**) but for risk score 2. **G** Same as (**D**) but for risk score 3. **H** LUSC risk score concordance: Same as (**D**) but for LUSC (*n* = 10 patients). Scale bars: (**A**–**C** + **E**–**G**) 3 mm (whole tumor sections) and 500 μm (tissue spots). Source data are provided as a Source data file.

enable cell-level registration. We limit the registration to basic geometric adjustments—specifically, affine transformations—to facilitate interpretability and avoid the generation of artifacts that would be infeasible to review manually with the amount of data available.

The quality of tissue alignment was scored computationally on a per-pixel level to allow rejection of areas where cells do not align well enough after registration, which can occur due to mechanical damage introduced between scanning phases. A subset of twenty spots containing misaligned regions was annotated to train and test the alignment mask quality control (QC) method proprietary to Aignostics. Spots with less than 0.7 square millimeters of aligned tissue were rejected for a lack of content. Spots with more than 3 square millimeters of aligned tissue were rejected from further analysis as the spots were broken into several shards that distort their spatial statistics.

## Statistics and reproducibility

Analyses were performed in Python 3.11.9 with pandas 2.3.2, numpy 2.3.2, scipy 1.16.2, lifelines 0.30.0, scikit-learn 1.7.2, and seaborn 0.13.2. Cellular graphics were prepared using Inkscape 1.3.2. In a first step, we extract the biological structure that is contained in the pixel space, which we call the image analysis pipeline. In a second step, the raw biological structures are summarized based on their spatial composition to create interpretable features. In a third step, we build predictive models for survival outcomes based on the spatial features.

## Step 1: Image analysis pipeline

**Tissue segmentation.** Although the tissue cores were punched from wider tumor regions, considering the whole spot as a tumor would be imprecise. In particular, necrosis and healthy tissue areas should be excluded when calculating the density of immune cells infiltrating the tumor. Therefore, we created a tissue segmentation model to distinguish carcinoma, stroma, necrosis and healthy areas. A total of 10871 tissue polygons were annotated, 697 for healthy, 4672 for carcinoma, 4571 for stroma and 931 for necrosis. We trained an ensemble of UNets[31] on a mixture of weighted cross entropy and DICE loss and collected the average F1 score obtained from a five-fold cross-validation, where the hold-out spots were selected based on the patient indicator.

**Cell detection.** We annotated 100,000 cell nucleus polygons to train and validate a StarDist model using the public repository version 0.8.2 commit id 466d0c on H&E-stained images at 20x resolution. The model employed the U-Net architecture and a star-convex polygon prediction framework, which we found to be particularly data-efficient. Recall and precision metrics were computed for 20 tissue spots, each exhaustively annotated over 100 μm² regions of interest.

**Cell classification.** A board-certified pathologist annotated a subset of the previously detected cells for marker positivity in each of the mIF channels separately. The labels were collected sparsely with an emphasis on spatial heterogeneity and local contrast of positive and negative pairs. We trained twelve independent binary models as they have to adjust to either attending to nuclear or membranous stains, have to consider the surrounding tissue contexts differently and have to suppress channel-specific artifact types and auto-fluorescence levels. To this end, we used a ConvNeXt Base[33] and provided it with a 256 square pixel crop centered on the cell to be classified at 10x magnification, together with the binary label from the pathologist. We only provided the H&E and respective mIF channel to classify, as preliminary experiments indicated a strong tendency for Clever Hans[34,88–90] overfitting if the model is presented with all mIF channels simultaneously. We repeated training on 5 folds to gather cross-validation scores and joined the models into an ensemble for inference on the whole cohort to further increase the performance for the following analysis. We distributed the inference of the 12 binary tasks with 3 ensemble members each applied to 53 million cells on a Kubernetes cluster, as required due to the total amount of 1.5 billion images.

**Co-registration of H&E and mIF images.** H&E and mIF staining were performed sequentially on the same section (for details, see "methods"). This is a critical prerequisite and allows for registration/alignment of the resulting images as well as the integration of model annotations on a single cell level. The H&E-based and mIF-based models, although optimized separately, were combined to work together as the H&E and mIF stainings were co-registered/spatially aligned (Fig. 6 provides an example of an alignment result between H&E and mIF). As a result, annotations generated by one algorithm in the H&E domain are seamlessly linked to annotations generated on the mIF domain. This alignment provides an integrated view of the TME, with H&E providing a robust basis for tissue segmentation and cell detection, and mIF adding detailed cell classification based on immuno-staining patterns.

**Motivation for training multiple binary classification models.** In the mIF domain, we trained individual binary classification models for each stain to optimize detection accuracies due to the unique patterns associated with each marker. Each stain required a different model (1) due to the different localization (nucleus, cytoplasm, membrane), which significantly affects the detection process, and (2) stain-specific imaging artifacts, such as speckle noise or blurred fluorescence, which must be accounted for to avoid false positives. During training, model performance was closely monitored on a hold-out validation set. This approach enabled us to achieve highly accurate results for each stain, ensuring that the detection and classification process was robust and adaptable to varying image conditions. These models were then applied to the whole cohort and offer improved performance compared to standard methods (see comparison with StarDist; Supplementary Fig. 4, Supplementary Table 1).

**Integration of H&E and mIF data.** Both the H&E and mIF models are the basis of the downstream analyses, including niche generation and survival prediction. The integration is required for two reasons. (1) The H&E-based model enables tissue-level segmentation into carcinoma, stroma, necrosis, and non-neoplastic regions, thereby ensuring that the analysis focuses on histologically relevant compartments. This kind of spatial annotation cannot be reliably obtained from mIF alone due to its limited morphological information. mIF, in turn, contributes precise cell-type classification through antigen-specific labeling, which is essential for phenotypic resolution. (2) The combined use of H&E and mIF is particularly critical for distinguishing morphologically similar cell types. For example, carcinoma cells and normal epithelial cells, such as pneumocytes or ciliated epithelial cells, often share expression of epithelial markers (e.g., cytokeratins), making them difficult to differentiate using mIF alone. Here, the morphological features captured by H&E—such as nuclear atypia, cell shape, and architectural arrangement—are used for accurate classification. Conversely, certain immune cell subtypes (e.g., B vs. T lymphocytes) are indistinguishable based on H&E but can be clearly resolved using mIF markers such as CD20 and CD3. Thus, the integration of both modalities provides complementary strengths: H&E delivers detailed tissue architecture and tumor morphology, while mIF enables high-resolution immunophenotyping. Together, they allow robust and context-aware cell classification across diverse tissue environments. A key feature of our approach is the cross-domain integration of these models for the characterization of cell niches. Cell niches are defined as distinct cellular neighborhoods within a 34 micron radius.

**Integration of all data modalities for survival analysis.** Survival analysis builds on these niche descriptions, incorporating them into a multivariate model alongside clinical metadata such as UICC staging. The detailed cellular composition of the niches, as generated by the integrated H&E and mIF models, provides a biologically meaningful context in combination with clinical metadata and is used to predict survival.

## Step 2: Extraction of spatial composition

**Marker density in tumor.** We computed the density of positive cells for each marker within the total tissue area and the tumor region, respectively. The total tissue was defined as the union of all tissue segmentation masks and the tissue alignment QC masks, effectively excluding only the white-background and damaged tissue. The tumor region was defined as the union of the tissue alignment QC mask and the tissue segmentation results for carcinoma and stroma. Measurements are scaled to cell counts per millimeter and provided as log1p-densities for any further numerical evaluation, as motivated by the strong right skew of the distribution. Further analyses comprised 2233 tumor samples from 663 patients (303 female, 360 male; mean age = 64.7 years; median age = 64.7 years; Fig. 2A, E) and 1657 tumor samples from 462 patients (116 female, 346 male; mean age = 67.8 years; median age = 68.7 years; Fig. 2B, G).

**Cell niches clustering.** While exact biological definitions of niche sizes do not exist, we oriented our niche size definition by the smallest and largest immune cells occurring in the TME[85,86]. In particular, we used the macrophage, for which a diameter of up to 30 µm has been documented[87] as the upper size limit. The lymphocyte, which typically has a minimal diameter of 8 µm, was used as the lower limit[87,89]. Considering the lymphocyte as the most frequent and smallest cell at the niche center, a 34 µm radius would allow for at least one adjacent macrophage, but also higher order neighbors, given the smaller lymphocyte sizes. This is consistent with parallel work using interaction ranges of 30–40 µm, such as Risom, Keren or Stoltzfus et al.[30,62,63].

Using the predefined radius of 34 µm, we used a scipy KDTree[91] to calculate the number of cells positive for each marker within this spatial neighborhood (see "Methods" section: *Cell classification*, page 33). These marker counts were logarithmically transformed to put more weight on the presence or absence of cells by gradually saturating the contribution, increasing count, to further put an emphasis on the relative composition of a neighborhood the expression values were divided by the local mass (percentage). To identify unique cellular neighborhoods, we performed Mini Batch K Means ++ clustering[92] with a batch size of 8000 to define 10 prototypical neighborhoods from the 53 million extracted histograms. Each cell was then assigned to the closest prototype using nearest-neighbor classification. We subsequently aggregated the data for each tissue spot to quantify how many cells corresponded to each identified niche. Further analyses comprised 2233 tumor samples from 663 patients (303 female, 360 male; mean age = 64.7 years; median age = 64.7 years; Figs. 3L, N, 5A) and 1657 tumor samples from 462 patients (116 female, 346 male; mean age = 67.8 years; median age = 68.7 years; Figs. 4L, N and 5B).

## Step 3: Predictive model

**Propensity score-matching.** The following motivates and documents the technical implementation of propensity score-matching, a method to balance patient characteristics between studies.

We carefully checked the compatibility of the training and validation cohorts before checking the transferability of the immunological characterization using a statistical pre-screening approach. This approach aimed to detect potential distributional shifts unrelated to the immunological characterization under investigation and thus aims to increase the reliability of our results.

First, we assessed cohort compatibility by training a classifier based on basic patient metadata, including UICC8 (integer number), observed overall survival (real number) and censoring event (Boolean value). The task of the classifier was to predict the origin of the patients (Berlin or Cologne). When the classifier could accurately predict a patient's location based on metadata alone, it indicated the presence of a particular patient stratum in one cohort but not in the other. Specifically, for LUAD cases, we obtained a non-chance F1 score indicating a distributional shift of key metadata between cohorts.

In response to this finding, we propose to remove unrelated cases from the training cohort (Berlin) while leaving the test cohort untouched. To achieve sample stratification, we use propensity score matching based solely on metadata, including UICC8, overall survival and censoring. Our approach uses a nearest neighbor-based algorithm for optimal complete matching. Specifically, we train a classifier to predict the treatment location (Berlin or Cologne) for each patient and use the model's logits to identify the most similar patient from Berlin for each Cologne patient. Importantly, we perform sampling without replacement to ensure that each patient is used only once.

## Maximum relevance minimum redundancy selection

We follow a maximum relevance minimum redundancy criterion[93] for feature selection, which we adapt for censored survival data. For each candidate feature, we fit a univariate Cox regression[92] and consider the achieved training set c-index as the feature's relevance. The feature with the highest relevance is picked first. The subsequent choices are guided by prioritizing high-relevance features but discouraging redundancy with respect to the already chosen features. To this end, we compute the cross-correlation between each candidate feature and the previously picked features and treat the absolute value of the average as a measure of redundancy. The candidate features are then sorted by the difference between the relevance score and the redundancy score. The first candidate is taken as the next choice, and the process repeats. Our implementation requires setting a maximum number of features to consider and is treated as a hyperparameter to be tuned in an inner fold cross-validation, for which we have integrated our implementation with sklearn[92].

## Maximum risk score approach

The maximum risk approach is motivated by the following reasoning: In conventional or AI-based biomarker studies or histopathological diagnostics to evaluate tumor aggressiveness, e.g., for histological grading and Ki67 immunohistochemistry, the evaluation of many biomarkers follows a so-called "hot-spot" strategy and does not use averages even if large resection specimens on whole slides are available[94–97]. For instance, tumor grading mostly does not rely on averages, but renders a diagnosis based on the presence of the features associated with the highest risk of disease progression. This means that a tumor is diagnosed as "high grade", even if the majority of the tumor shows "low grade" morphology and only a subregion shows "high grade" features.

We used maximum risk scores, because the purpose of our approach was to identify patients having an increased risk with high sensitivity in order to ensure that the niche-based patient stratification will not lead to withholding a necessary adjuvant therapy.

## Survival analysis

The curation of a representative and homogenous patient cohort is crucial for survival analysis, especially in observational studies, to ensure that main effects are not biased by distributional shifts within the cohort. In our main results, we focused on the analysis of early-stage patients, with a separate analysis for late-stage (UICC8 = 4) to be found in the Appendix. We also remove patients with r-status = 1, meaning that patients for whom complete tumor removal could not be confirmed. In addition, we removed patients whose observed or

censored survival was less than three months, as their death can also be attributed to complications during or after surgery, limiting their usefulness for gaining insights into cancer immunology. Another notable distributional shift may result from treatment with adjuvant therapy, which we therefore present in a separate analysis in the Appendix. The patient numbers in Fig. 1A reflect the selection criteria introduced here.

Patients are represented by up to four tissue spots. We analyzed them using a multiple instance learning approach and chose a late fusion strategy for better interpretability. In this approach, each spot was evaluated individually for survival based on its niche composition features, and the model outputs a continuous risk score (logit) for each spot. To derive a patient-level risk estimate, we applied max-pooling across all spot-level outputs—meaning the highest risk score among a patient's spots determines the overall risk classification ("hot-spot approach"). This reflects the clinical rationale that the most aggressive region within the tumor is most prognostically relevant. The final patient-level prediction is thus determined by the maximum risk score across all available spots.

The cellomics features are typically right-skewed with a heavy tail, which motivates the use of a non-linear normalization transform to better comply with the assumption of the following regression modeling. Instead of employing a fixed log1p transformation, as is often used in omics pipelines, we opt for the power transformation of yeo-johnson[98] on the respective training for the increased flexibility. Inner fold cross-validation was used to tune sweep over a number of features to select, and a range of alpha regularization and select the best combination based on the average multiple-instance c-index. To assess the improvement in outcome prediction of the cell niche model compared to the clinical baseline (UICC8), we computed the relative improvement in c-index, accounting for its informative range between 0.5 (random prediction) and 1.0 (perfect prediction). It is defined as follows:

$$\text{Relative improvement} = \frac{c - model - 0.5}{c - baseline - 0.5} \qquad \text{(Eq.1)}$$

where c-model denotes the c-index of the cell niche model, c-baseline is the c-index of the clinical baseline (UICC8), and 0.5 represents the lower bound of the c-index, corresponding to random prediction.

The cox regression trained on Berlin is then applied to the patients from Cologne. We collected the risk attributions generated by the Cox regression model and stratified the test cohort into three groups based on the distribution of these scores. Specifically, we divided the patients into tertiles corresponding to the lower, middle, and upper thirds of predicted risk. These three groups are referred to in the main text as risk strata RS1 (low risk), RS2 (intermediate risk), and RS3 (high risk). This post-hoc discretization was introduced to enhance the interpretability of the survival analysis and visualization, while maintaining the model's continuous output during training and inference. The use of three risk groups provides a natural counterpart to the UICC clinical staging system for early stage lung cancer (stages I–III), according to which patients in our study were classified. This alignment facilitates integration with established staging frameworks and enhances clinical interpretability. Stratifying patients into low, intermediate, and high-risk groups is also a widely used and practical approach across cancer types, including lung, prostate, and breast cancer. Compared to a simple low/high split, the three-tiered system offers more nuanced risk assessment while remaining actionable in routine care and clinical trial design. Kaplan-Meier curves are tested for statistically significant separation among the patient strata using log-rank tests[99]. Demographic details of the analyzed patient subsets are as follows: Fig. 6A–C comprises 109 patients (45 female, 64 male; mean age = 64.7 years; median age = 65.4 years). Figure 6D includes RS1

($n = 61$; 29 female, 32 male; mean = 64.3; median = 64.0), RS2 + 3 ($n = 163$; 64 female, 99 male; mean = 65.8; median = 66.1), and UICC I ($n = 148$; 66 female, 82 male; mean = 65.2; median = 64.8). Figure 7A–C represent 109 patients (24 female, 85 male; mean age = 69.9 years; median age = 70.4 years), and Fig. 7D includes RS1 ($n = 66$; 20 female, 46 male; mean = 69.0; median = 69.9), RS2 + 3 ($n = 209$; 54 female, 155 male; mean = 69.3; median = 70.3), and UICC I ($n = 163$; 49 female, 114 male; mean = 69.3; median = 70.2). The corresponding Supplementary analyses comprise: Supplementary Fig. 12A, B (109 patients; 45 female, 64 male; mean = 64.7; median = 65.4), Supplementary Fig. 13A, B (109 patients; 24 female, 85 male; mean = 69.9; median = 70.4), Supplementary Fig. 14A ($n = 72$; 39 female, 33 male; mean = 62.5; median = 61.6), Supplementary Fig. 14B ($n = 18$; 6 female, 12 male; mean = 68.0; median = 68.8), Supplementary Fig. 15A ($n = 164$; 75 female, 89 male; mean = 64.4; median = 64.1), and Supplementary Fig. 15B ($n = 98$; 18 female, 80 male; mean = 64.1; median = 63.6).

**Immunology phenotype loading.** The marker panel was selected with the aim to assess in addition to the cancer cells the distribution of the main immune cell populations (e.g., T cells, TAMs, B cells, and NK cells) according to established lineage markers (e.g., CK, CD3, CD68, CD20, CD56, CD4, CD8), which have been described to be present in the TME of LUAD and LUSC[22]. The remaining markers were selected to determine the activation state and the cytotoxic or immunosuppressive potential of the cells (e.g., PD-1, FoxP3, PD-L1, CD163).

A complete analysis of all $2^{12} = 4096$ theoretically possible combinations from the 12 selected markers would not be meaningful since many marker combinations are not present physiologically. For this reason, we set up a table with phenotype definitions (Supplementary Data 2), in which we firstly list the main cell populations. Secondly, we subcategorized the main populations in well described immune cell subpopulations (such as CD4 + , CD8 + , and regulatory (FoxP3 + ) T cells or TAM1 and TAM2). We next used the markers granzyme B and PD-1 to allocate lymphocytes as activated. With the expression of PD-1 only, we classified the lymphocytes as exhausted. PD-L1 expression—independent of the cell lineage—was used to subclassify cells as immunosuppressive. No additional subpopulations were defined from combinations of those cell populations and markers whose biological relevance is currently unknown, e.g., the expression of CD4 and/ or CD8 expression in TAMs.

For easier interpretability, the aim of our classification was also to avoid populations without known biological roles. Furthermore, we attempted to limit the fragmentation of our classification because too many populations would increase the risk of overfitting the machine learning model. This approach yielded 23 medium-grained or 43 fine-grained cell phenotypes. Examples of all 43 fine-grained cell phenotypes along their marker expression are shown in Supplementary Fig. 3.

**LUAD and LUSC, matched niches based on phenotyping**
We employ partial optimal transport[64,65] to quantitatively determine matched pairs among the independently derived niches in LUAD and LUSC. The niches are characterized by the distribution of phenotypes they encompass. To measure the similarity of categorical distributions, the chi-squared test is often used, and serves us as a distance measure for the identification of analogous niches. We adopt the symmetric distance formula from the chi-squared kernel[100]

$$\sum_i \frac{(x_i - y_i)^2}{(x_i + y_i)} \qquad \text{(Eq.2)}$$

Since the number of matching pairs is not known a priori, we systematically evaluated all scenarios ranging from zero matches to full matches (10/10). Following best practices in clustering and dimensionality reduction, we assume an "elbow" phenomenon that

describes the transition from easily matched pairs to more difficult matches. Therefore, we analyze the log transport costs to identify this characteristic pattern.

### Step 4 – Orthogonal validation

**Cell classification—in-house optimized versus standard StarDist.** We compared the performance of our model against publicly available baseline checkpoints, referred to as the "Standard StarDist H&E model" and the "standard StarDist DAPI model." For each model, F1-scores were calculated on the 20 test tissue core regions of interest. Differences in average F1-scores were assessed for statistical significance using a one-sided paired-sample permutation test with 9999 iterations.

**Immunohistochemical based validation of mIF derived PD-L1 scoring.** We validate the mIF based quantification of the PD-L1 state with an orthogonal protocol based on immuno- histochemistry brightfield microscopy and manual scoring from expert pathologists. The corresponding patient subset comprised $n = 769$ (301 female, 468 male; mean age = 65.7 years; median age = 66.3 years).

For this, TMA blocks were sectioned at a thickness of 4 µm for immunohistochemical analysis. The sections were incubated in CC1 mild buffer (Ventana Medical Systems, Tucson, AZ, USA) at 100 °C for 30 min. Afterward, they were treated with an anti-PD-L1 antibody (E1L3N, Cell Signaling, #13684S, 1:200) at room temperature for 60 min and visualized using the avidin–biotin complex method with DAB staining. The BenchMark XT immunostainer (Ventana Medical Systems, Tucson, AZ, USA) was used for these steps. Cell nuclei were counterstained with hematoxylin and bluing reagent (Ventana Medical Systems, Tucson, AZ) for 12 min.

The stained sections were analyzed under an Olympus BX50 microscope (Olympus Europe), and the tumor proportion score (TPS) was calculated as the percentage of PD-L1-positive tumor cells among all viable tumor cells. TPS scoring was performed independently by two pathologists (S.S. and F.K.), who have 8 and 14 years of experience in lung pathology, respectively. Digital histological images were captured using the PANNORAMIC 1000 slide scanner (3DHISTECH).

To assess the precision of the AI-powered approach in identifying prognostically relevant immune states within the TME, a correlation analysis was performed between the TPS scores obtained through multiplex immunofluorescence (mIF) by the AI approach and those determined via immunohistochemistry (IHC) by the pathologists (Supplementary Fig. 6). The correlation was tested for statistical significance using scipy.stats.pearson_r.

**H&E based validation of mIF derived lymphocyte counts.** We validated the multiplex immunofluorescence (mIF)-based quantification of lymphocytes by comparing it with an independent H&E-based quantification using a state-of-the-art foundation model. The corresponding patient subset comprised $n = 1125$ (419 female, 706 male; mean age = 66.0 years; median age = 66.3 years). For this purpose, we employ an in-house variant of RudolfV[101], fine-tuned for cell classification. The number of lymphocytes detected is compared to the number of CD3- or CD20-positive cells identified in mIF on a tissue core-by-tissue core basis (Supplementary Fig. 7).

Given the heavily right-skewed distribution of cell counts per tissue core, the data are log-transformed prior to analysis. This transformation normalizes the distribution and ensures the validity of the subsequent statistical test. Statistical significance of the correlation between the two methods is assessed using Pearson's correlation coefficient, implemented via scipy.stats.pearson_r.

**Risk score heterogeneity and resection based validation of TMA derived risk quantification.** To evaluate the adequacy of TMAs for risk score assessment, we analyzed inter- and intra-tumoral heterogeneity of niche-based risk scores across tissue cores from each patient and throughout the cohort. The analysis included 2233 tumor samples from 663 LUAD patients (303 female, 360 male; mean age = 64.7 years; median age = 64.7 years) (Supplementary Fig. 20 A, C, D, F) and 1657 tumor samples from 462 LUSC patients (116 female, 346 male; mean age = 67.8 years; median age = 68.7 years) (Supplementary Fig. 20B, C, E, G).

We further evaluated the reliability of TMA cores as surrogates for whole tumor sections by comparing case risks derived from surgical whole tissue sections with those from their corresponding TMA cores. Each resection is processed through the same pipeline, comprising registration, tissue segmentation, cell detection, cell classification, neighborhood aggregation, and cell-niche assignment. The analyses were conducted on two representative subsets of the main NSCLC cohort. Figure 8D and Supplementary Fig. 23A, B include $n = 10$ patients (6 female, 4 male; mean age = 68.1 years; median age = 68.0 years), while Fig. 8H and Supplementary Fig. 23C, D include another $n = 10$ patients (3 female, 7 male; mean age = 69.8 years; median age = 74.0 years).

The segmentation model subsets the resection to tumor-stroma microenvironment regions, excluding other tissue types such as healthy, necrotic, or fatty tissue. The risk model, which links UICC8 staging and niche composition to tissue-core risk scores, is applied to the entire tumor region on a sliding window basis. Specifically, virtual tissue cores are placed to exhaustively cover the tumor area, and risk scores are aggregated across all virtual cores. The case risk for whole tissue sections, as for TMA cores, is defined as the maximum risk score among all virtual cores.

To assess reliability, we evaluate whether the risk conclusions derived from virtual cores of a resection align with those from the subset of TMA tissue cores. Concordance between whole resection-based and TMA core-based risk evaluations is tested for statistical significance using Pearson's correlation coefficient, implemented via scipy.stats.pearson_r.

**Association of risk scores with known mutations and covariates.** We obtained panel sequencing data (18 genes) for a subset of LUAD- and LUSC-patients (Supplementary Fig. 16A). The LUAD subset comprised $n = 147$ patients (63 female, 84 male; mean age = 65.4 years; median age = 65.9 years), and the LUSC subset comprised $n = 178$ patients (42 female, 136 male; mean age = 69.6 years; median age = 71.0 years; Supplementary Data 7). Statistical significance was evaluated by constructing contingency tables for each gene, with rows representing mutation status (mutated vs. wild type) and columns representing risk categories (e.g., low, mid, high). A chi-squared test was then applied to each table. Our analysis showed no significant correlation between the derived risk scores and mutations in any of the frequently mutated genes in LUAD (Supplementary Fig. 16B) or LUSC (Supplementary Fig. 16B).

Furthermore, it has been demonstrated that additional covariates, including sex, age, ECOG performance status, smoking status, tumor grade, thyroid transcription factor 1 (TTF-1) expression, and growth pattern, are associated with patient survival in early-stage NSCLC. We therefore conducted an analysis to determine whether there were any potential correlations between TME risk profiles and these covariates. Our analysis showed no significant correlation between the derived risk scores and covariates (Supplementary Fig. 17). The analysis comprised the following patient subsets: Supplementary Fig. 17A, B ($n = 224$; 93 female, 131 male; mean age = 65.4 years; median = 65.7), 17 C ($n = 194$; 85 female, 109 male; mean = 65.1; median = 65.4), 17D ($n = 131$; 58 female, 73 male; mean = 65.8; median = 65.9), 17E ($n = 211$; 88 female, 123 male; mean = 65.7; median = 65.7), 17 F ($n = 180$; 70 female, 110 male; mean = 65.4; median = 65.8), 17 G ($n = 216$; 90 female, 126 male; mean = 65.5; median = 65.7), 17H + I (n = 275; 74 female, 201 male; mean = 69.2; median = 70.3), 17 J ($n = 234$; 61 female, 173 male;

mean = 69.3; median = 70.5), 17 K (n = 156; 46 female, 110 male; mean = 68.6; median = 69.9), 17 L (n = 256; 67 female, 189 male; mean = 69.3; median = 70.3), and 17 M (n = 270; 71 female, 199 male; mean = 69.2; median = 70.3).

**Association of risk scores with fibroblast abundance.** Fibroblasts were detected in H&E-stained images using the RudolfV[101] pathology foundation model. The model was applied to all tissue spots in the LUAD (2189 tumor samples from 663 patients (303 female, 360 male; mean age = 64.7 years; median age = 64.7 years; Supplementary Fig. 18C–F) and LUSC (1618 tumor samples from 462 patients (116 female, 346 male; mean age = 67.8 years; median age = 68.7 years; Supplementary Fig. 19C–F) cohorts. For each patient, fibroblast abundance was quantified per spatial niche both as absolute counts and as proportions relative to all cells within the niche. Correlations between fibroblast abundance and niche frequencies were computed using Pearson's correlation coefficient. Differences in fibroblast abundance between "hot" and "cold" niches were assessed using a Chi-squared test. To evaluate prognostic associations, patients were stratified into tertiles based on either their fibroblast maximum abundance over tissue spots (max pooling) or their fibroblast mean abundance over tissue spots, analogous to the risk score analyses for niche patterns. Survival differences between strata were analysed using Kaplan–Meier curves and log-rank tests. The LUAD survival cohort comprised n = 108 patients (44 female, 64 male; mean age = 64.7 years; median age = 65.4 years; Supplementary Fig. 18G. H), and the LUSC survival cohort comprised n = 109 patients (24 female, 85 male; mean age = 69.9 years; median age = 70.4 years; Supplementary Fig. 19 G + H).

### Step 5 – Visualizations

**Feature distributions.** All cluster maps were z-scored over columns (features), hierarchical agglomerative clustering was applied with Euclidean distance and single linkage. For better visual contrast, the colormaps were non-linearly compressed with a logarithm, including a multiplicative saturation factor if necessary.

Stacked histograms were sorted based on the one-dimensional projection of their kernel principal components[102], done separately for rows and columns. We use a Gaussian kernel with a length scale of one over the number of features. This choice was made to better convey the idea of the multi-dimensional cumulative distribution of the cohort.

**The immunology atlas.** We embed the tissue spots in a two-dimensional plane to give a comprehensive overview of the concentration and spread among the immuno-composition we found via application of the niches. The location of each spot is determined by the loadings of cell niches measured in each spot. We used a UMAP[103] projection with the number of neighbors set to 100, minimum distance set to 8, and a spread of 8. This was chosen to avoid dense clustering that would have resulted in overlapping tissue composition thumbnails. Each tissue thumbnail comes with a circular bar plot that summarizes the niche concentration of each spot. See Figs. 3 and 4.

### Ethics declaration

The study was performed according to the ethical principles for medical research of the Declaration of Helsinki and was approved by the Ethics Committee of the Charité University Medical Department in Berlin (EA4/082/22). All patients provided written informed consent for the scientific use of their archived tissue and associated clinical data. The study was retrospective and involved no additional procedures or interventions; therefore, no participant compensation was provided. Information on the race, ethnicity and socioeconomic status of patients was not available. Mutation status, recurrence information and smoking status were incomplete at the time of writing.

### Reporting summary

Further information on research design is available in the Nature Portfolio Reporting Summary linked to this article.

## Data availability

All processed cell-classification results and niche annotations have been deposited in Zenodo under accession code https://doi.org/10.5281/zenodo.11395885. High-resolution image tiles for each patient, including all multiplex immunofluorescence (mIF) channels and H&E (14,000 images), have been deposited in Zenodo under accession code https://doi.org/10.5281/zenodo.16882468. The complete raw dataset (>1 TB), including patient metadata and full image files, is available under restricted access due to size and privacy considerations. Access can be requested from the corresponding author and will be granted within one week upon submission of a signed non-commercial use declaration and verification of an academic email address. All bona fide academic requests will be approved. Source data for reproducing the figures are provided with this paper. No publicly available datasets were reused in this study. Source data are provided with this paper.

## Code availability

Code is available at https://github.com/gabrieldernbach/cell_niches.

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

## Acknowledgements

Funding was provided through BIFOLD/BMBF. This work was partly funded by the German Ministry for Education and Research (under refs. 16 LW0239K, 01IS14013A-E, 01GQ1115, 01GQ0850, 01IS18056A, 01IS18025A and 01IS18037A) and BBDC/BZML and BIFOLD. Furthermore, KRM was partly supported by the Institute of Information & Communications Technology Planning & Evaluation (IITP) grants funded by the Korea Government (MSIT) (No. 2019-0-00079, Artificial Intelligence Graduate School Program, Korea University and No. 2022-0-00984, Development of Artificial Intelligence Technology for Personalized Plug-and-Play Explanation and Verification of Explanation). M.P.D. was supported by the Berlin Institute of Health (Clinician Scientist Program) and DKTK Berlin (Young Investigator Grant 2022). The authors thank Dzmitry Talkach, Fabian Spieß, Moritz Krügener, Andreas Kunft and Jesús Marín González for help with engineering and graphical design.

## Author contributions

Conceptualization: F.K., G.D., K.R.M., L.R., M.A., S.R., and S.S.; Data curation: G.D., S.R., S.S.; Formal analysis: G.D., K.S., M.F., N.F., and S.S.; Investigation: G.D., K.S., L.R., S.G., and S.S.; Methodology: G.D., K.S., L.R., M.A., and S.S.; Resources: A.Q., A.V., D.H., G.S., J.C.R., K.B., K.H., M.H., R.B., S.M., S.S., U.K., and W.R.; Software: C.B., G.D., K.S., and S.G.; Validation: G.D., L.R., M.A., M.F., N.F., and S.S.; Writing—original draft: F.K., G.D., K.R.M., L.R., M.A., N.F., P.G.K., and S.S.; Writing—review and editing: C.F., F.K., G.D., I.K.N., K.R.M., L.R., M.A., M.F., M.P.D., N.F., P.J., and S.S.

## Funding

## Competing interests

F.K., M.A., and K.R.M. are co-founders of Aignostics. F.K. is lead medical advisor and board member of Aignostics. K.R.M. is lead technical advisor to Aignostics. M.A. is CTO of Aignostics. G.D. and S.S. are part-time employees at Aignostics. DH is a member of the scientific advisory board at Aignostics. The other authors declare no conflict of interest.

## Additional information

¹Institute of Pathology, Charité - Universitätsmedizin Berlin, corporate member of Freie Universität Berlin, Humboldt-Universität zu Berlin and Berlin Institute of Health, Berlin, Germany. ²Aignostics, Berlin, Germany. ³Institute of Pathology, Ludwig-Maximilians-Universität München, Munich, Germany. ⁴Department of Hematology, Oncology, and Tumor Immunology, Charité - Universitätsmedizin Berlin, corporate member of Freie Universität Berlin and Humboldt-Universität zu Berlin, Berlin, Germany. ⁵BIH Center for Regenerative Therapies (BCRT), Berlin Institute of Health at Charité - Universitätsmedizin Berlin, Berlin, Germany. ⁶German Cancer Consortium (DKTK), partner site Berlin, and German Cancer Research Center (DKFZ), Heidelberg, Germany. ⁷Berlin Institute of Health (BIH), Berlin, Germany. ⁸BIFOLD - Berlin Institute for the Foundations of Learning and Data, Berlin, Germany. ⁹Max-Delbrück-Center for Molecular Medicine in the Helmholtz Association (MDC), Proteomics Platform, Berlin, Germany. ¹⁰ECRC Experimental and Clinical Research Center, Charite - Universitätsmedizin Berlin, corporate member of Freie Universität Berlin and Humboldt Universität zu Berlin, Berlin, Germany. ¹¹Institute of Pathology, University Hospital Cologne, Köln, Germany. ¹²Department of Infectious Diseases and Respiratory Medicine, Charité - Universitätsmedizin Berlin, corporate member of Freie Universität Berlin, Humboldt-Universität zu Berlin and Berlin Institute of Health, Berlin, Germany. ¹³Bethanien Hospital, Clinic of Pneumology and Allergology, Center for Sleep Medicine and Respiratory Care, Institute of Pneumology at the University of Cologne, Solingen, Germany. ¹⁴Department of Cardiothoracic Surgery,

University Hospital Cologne, Solingen, Germany. [15]Charité Comprehensive Cancer Center, Charité - Universitätsmedizin Berlin, corporate member of Freie Universität Berlin, Humboldt-Universität zu Berlin and Berlin Institute of Health, Berlin, Germany. [16]Department of General, Visceral, Vascular and Thoracic Surgery, Charité - Universitätsmedizin Berlin, Berlin, Germany. [17]Ultivue, Berlin, Germany. [18]Machine Learning Group, Technische Universität Berlin, Berlin, Germany. [19]Max-Planck-Institute for Informatics, Saarbrücken, Germany. [20]Department of Artificial Intelligence, Korea University, Seoul, South Korea. [21]German Cancer Consortium (DKTK), partner site Munich, and German Cancer Research Center (DKFZ), Heidelberg, Germany. [22]Bavarian Cancer Research Center (BZKF), Munich, Germany. [23]These authors contributed equally: Simon Schallenberg, Gabriel Dernbach. ✉e-mail: klaus-robert.mueller@tu-berlin.de; frederick.klauschen@med.uni-muenchen.de

