## [Transparent Peer review file · Nature Communications]

AI-powered spatial cell phenomics enhances risk stratification in non-small cell lung cancer

Corresponding Author: Professor Frederick Klauschen

Version 0:

Reviewer comments:

Reviewer #1

(Remarks to the Author)

The authors use data from a large retrospective surgical cohort to identify spatial relationships between cells of the immune microenvironment in non-small cell lung cancer. They characterise recurrent cellular neighbourhoods, independently in lung adenocarcinoma (LUAD) and lung squamous cell carcinoma (LUSC) and use this information in order to stratify patients better for treatment in the adjuvant setting. Notably, this is additive to stage and the risk profile can be calculated from small tissue fragments, which have implications for diagnostic biopsy specimens.

Main points:

What was the motivation to discover cell niches independently in LUAD and LUSC, if the plan was to unite these post-hoc? Would doing niche discovery across the two cancer subtypes make it easier to compare the relative frequency of niches? In LUAD specifically, were there any inclusion/exclusion criteria? Were all growth patterns represented, including those without stromal invasion (lepidic, papillary, conventional micropapillary)?

Was there any relationship between growth pattern and cell niches? - the needs to be explored as growth pattern is so determinative of the tumour microenvironment.

Figures throughout would benefit from slightly better readability - larger text, inclusion of colour bars in heatmaps to define colour scales, annotations to highlight key results, axes labels and titles

Smaller points moving through the text:

In the introduction, some more detail about the TME in NSCLC would better set the stage for the rest of the article

Could fig 1 be a supplementary figure? (From panel B onwards)

Fig 2e, g what are the un-numbered groups?

For clarity, I'd recommend using different notation to refer to groups of cell niches (as in Fig 3k/4k) and groups of patients (as in Fig 3N/4N)

In Fig 3n, Fig 4n, some cases are not assigned a group (no roman numeral on the heatmap). Why is that?

Throughout page 14-17, there are many comparisons made between LUAD and LUSC but these are not backed by statistical testing

How were the risk scores calculated and what was the rationale for 3 risk group categories?

(Remarks on code availability)

Reviewer #2

(Remarks to the Author)

NCOMMS-25-40035

"AI-powered spatial cell phenomics enhances risk stratification in non-small cell lung cancer"

The authors study resected tumor specimens from 1,168 non-small cell lung cancer (NSCLC) patients, divided between lung adenocarcinoma (LUAD) and lung squamous carcinomas (LUSC) that came from two independent German cancer centers (Berlin and Cologne). They obtain H&E stained images of the tumors and also images of immunohistochemical stains (IHC) for 12 immune cell related markers and developed computational biology "AI-powered spatial cellomics approach"

approaches to analyze these image data sets and classify tumors, and the tumor microenvironment including the identification of “niches”. In addition, they determined the spatial relationship of tumor cells to the TME niches. All of these image data were computationally sorted into groups (“clusters”) and also related to survival data. Their image analyses improved survival annotation by some 14% for LUAD and 47% for LUSCs. They conclude: “ Our results showed that complex immune cell niche patterns identify potentially undertreated high-risk patients qualifying for adjuvant therapy. Our approach highlights the potential of AI-powered multiplex imaging analyses to better understand the contribution of the TME to cancer progression and to improve risk stratification and treatment selection in NSCLC. “

Comments to the authors:

This manuscript is reviewed in the setting of the urgent need to use image analysis and computational biology approaches on routine pathologic materials, as well as IHC stains, to classify lung tumors in a manner that would enhance the selection of the best therapy and obtain prognostic information for individual patients. In addition, to be able to independently validate the authors’ approaches and results, compare these to multiple other approaches, and to be able to use the large datasets for further research outside of the authors’ domains.

1. The amount of work is very large, clearly presented, and appears to be carefully done. The authors are to be commended on the scale and detailed analyses they have performed.
2. The ability of other investigators to obtain their image and clinical datasets easily for both independent validation, and to use for comparison to other datasets is very important. I believe that is the authors’ intent as well, but it will be important for this to be clearly stated and the conditions and methods for obtaining these key data spelled out. With such access the value and reason for publishing this paper is greatly enhanced.
3. There have been a large number of other publications dealing with various computational approaches to image analyses of lung cancers and other tumor types. While it is impractical to try to compare their bespoke approach to a large number of these, it would be useful to know if the authors took a few of the most prominent other publications dealing with identification of the tumor microenvironment from H&E images and compared them to their own approach. Are the various groups and niches identified by the different approaches similar or quite distinct? Is there any value of combining the approaches and subsequent classifications the authors developed with classes identified by other approaches?
4. A key aspect of course would be to integrate current widely used in standard clinical pathology IHC stain assessments of NSCLCs with the authors’ image analyses such as TTF-1, Napsin A, CK5/6, p63, p40, synaptophysin, chromogranin A, CD56. It would seem likely that information on some of these routinely used IHC markers for NSCLC diagnosis are available and if so, it would be very helpful to see how this information integrated with the authors’ image analyses classes and niches.
5. Likewise, my guess is that many of these cases had routine oncogenotypes obtained as part of standard of care, such as EGFR, KRAS, TP53, STK11/LKB1 and others. If so it would be helpful to see how these are integrated and compare to the authors’ classifications.
6. I did not find a clear statement about analysis of some standard clinical demographic data with the classifications including age, gender, smoking status, race. I presume these analyses were done and if they did not impact on the classification (or if they did) this should be stated. Likewise as part of the deposited datasets available for other investigators to use, it will be important to also have some basic clinical demographic data such as these available as well.

(Remarks on code availability)

Reviewer #3

(Remarks to the Author)

The authors use two large cohorts for training and testing of a model that predicts the clinical outcomes of lung cancer patients. Adeno- and squamous lung cancers are analyzed separately for 10 cell niches with a circle with a diameter of 34 microns, i.e. the size of a macrophage, in the TME. The niches are divided into hot and cold immune cell regions based on lymphocyte content using 12 antibodies in a cyclic IF staining protocol followed by an H&E stain. Adding the niches to the TNM classification improves the survival prediction. Most importantly, the niches identify TNM stage-I cases who would benefit from adjuvant chemotherapy. Altogether, adding niche information increases the c-index above the TNM UICC8 baseline.

Workflow: pixel level registered 12-plex IF and H&E tiles from TMA blocks are used to train a risk score predictor. Each case is represented by 4 cores in the TMA.

The pipeline consists of the following steps:

Stage 1: tissue segmentation into cancer, stroma, benign, necrosis. A U-Net model trained in house possesses an average F1-score = 0.92

Stage 2: cell identification. Nuclear segmentation was performed with an optimized StarDist model, providing an F1-score = 0.91

Stage 3: cell classification. For cell classification 12 independent ConvNet models were trained, using each mIF channel and the corresponding H&E as the input. The patch level F1-score = 0.91)

Stage 4: niche identification (K-mean clustering of the 34µm diameter neighborhoods for each cell; k is set to 10)

Stage 5: survival analysis: the niche composition of each core is used to predict patient survival using a multiple instance learning model. The highest risk core is used together with clinical variables is used to train a Cox regression model in the training cohort. The model is applied to the test cohort.

- Will the work be of significance to the field and related fields? How does it compare to the established literature? If the work is not original, please provide relevant references.

1. This work is significant to the field and related fields with regards to the analysis of the tumor microenvironment. This interesting study uses a large lung cancer cohort with high quality data. The main contribution to the field is that the analysis pipeline is straight forward and easy to recreate at a new local site.

2. Adding the results from the niche analysis to the TNM staging has the potential to improve the stratification for adjuvant therapy.

3. multiple consideration related to data structure and quality are addressed using the appropriate statistical methods, leading to rigorous data analysis.

4. The cell types depicted in ST2 demonstrate how the 12-plex is used to derive almost 200 different subpopulations of immune cells, which leads to the cluster composition and cell composition of clusters shown in the supplementary tables. This data is a significant contribution to the field.

- Does the work support the conclusions and claims, or is additional evidence needed? The conclusions are supported by the data.

- Are there any flaws in the data analysis, interpretation and conclusions? - Do these prohibit publication or require revision? The major concerns are:

1. The immunosuppressive cancer phenotype could be caused by cells in the stroma, for example fibroblasts. Fibroblasts are not considered in the current study. Can the H&E images be used to determine the amount of fibroblasts in the TME and a relationship between numbers of fibroblasts and niche composition. Please consider the effects of fibroblasts as described in "Cords, L., Engler, S., Haberecker, M., Rüschoff, J.H., Moch, H., de Souza, N., and Bodenmiller, B. (2024). Cancer-associated fibroblast phenotypes are associated with patient outcome in non-small cell lung cancer. *Cancer Cell*" in the discussion of the paper.

2. The risk score stratification using niche variables improves the baseline c-index calculated based on UICC8 TNM variables only by a small amount in terms of c-index (proportion of patient pairs in which the one that has higher risk score experienced death sooner than the lower score one). However, the improvement compared to the baseline TNM UICC8 is small: e.g. the baseline c-index for LUSC is 0.630; and the model achieved a c-index of 0.692. It is unclear how the authors calculated the 47% improvement in the c-index that they claim.

3. lack of clarity of several steps in the risk score determination:

a. it is unclear how the ConvNet prediction of 256 x 256 tiles will be processed to arrive at the niche classification.

b. Aggregation of niches for survival analysis uses a MIL framework. What is the input and output of the MIL ?

c. It is not clear how RS1, RS2, and RS3 are labeled for training of the Cox regression model for survival prediction.

4. Cell classification: if the algorithm learns features from H&E based on IF annotations, why does it need the IF during inference ? What features does the algorithm learn ?

5. Questions / comments related to figures:

a. Figure 1. Does the composition of the immune cell infiltrate change with tumor cell content ? What are the purple cell populations in panel A ? Please use the same color for subpopulations of immune cells in panel A and B.

b. Figure 1: C. poor H&E quality and green artifacts. Please improve quality of images.

c. Figure 1: E and G: these panels are not informative because many cell types show no difference between clusters. Show cell phenotypes that drive the clustering and that separate the clusters.

Minor concerns:

The classification of carcinomas by immune status refers to references between 2011 and 2026 (Schalper et al. 2015; Mahmoud et al. 2011 and Taube et al. 2012).

- Is the methodology sound? Does the work meet the expected standards in your field? The methodology is sound.

- Is there enough detail provided in the methods for the work to be reproduced? Please see major concerns for questions that need to be answered to reproduce this work.

(Remarks on code availability)

Reviewer #4

(Remarks to the Author)

(Remarks on code availability)

The repository appears to be well structured. The README file is comprehensive. The code is executable according to the provided instructions, and results obtained. However, the results are based on a subset of the full data set used for the paper. The full data is described as "available upon request", however it is not sure if all requests will be granted, or there will be additional criteria for granting access.

Reviewer #5

(Remarks to the Author)

(Remarks on code availability)

Version 1:

Reviewer comments:

Reviewer #1

(Remarks to the Author)

Thank you for comprehensively meeting our inquiries and requests.

(Remarks on code availability)

Reviewer #3

(Remarks to the Author)

We appreciate the thorough responses by the authors to our concerns, in particular to those that were overlapping with one of the other reviewers. Both, additional experimental data and information in additional figures and clarifications in the text are helpful. We don't have any additional questions or concerns that need to be addressed.

(Remarks on code availability)

Reviewer #5

(Remarks to the Author)

(Remarks on code availability)

Response to Reviewers

(Schallenberg & Dernbach et al.; *NCOMMS-25-40035*)

Reviewers' comments are in *italics* highlighted in *purple*, references to figures in the revised manuscript or supplement are highlighted in *blue*. Answers to reviewers' comments and references to rebuttal figures are black.

Reviewer #1

The authors use data from a large retrospective surgical cohort to identify spatial relationships between cells of the immune microenvironment in non-small cell lung cancer. They characterise recurrent cellular neighbourhoods, independently in lung adenocarcinoma (LUAD) and lung squamous cell carcinoma (LUSC) and use this information in order to stratify patients better for treatment in the adjuvant setting. Notably, this is additive to stage and the risk profile can be calculated from small tissue fragments, which have implications for diagnostic biopsy specimens.

Main points:

- 1. What was the motivation to discover cell niches independently in LUAD and LUSC, if the plan was to unite these post-hoc? Would doing niche discovery across the two cancer subtypes make it easier to compare the relative frequency of niches?*

We apologize if we made the impression to unite LUAD and LUSC post-hoc, which was not our intention. We performed separate analyses for LUAD and LUSC, because of clinicopathological and methodological aspects.

Clinicopathological rationale

LUAD and LUSC are increasingly recognized as distinct tumor entities, owing to biological, histological, and clinical differences. The key distinctions include the following:

- **Etiology:** LUAD occurs more frequently in non-smokers and females, while LUSC is strongly associated with smoking (Wakelee et al., 2007).
- **Anatomical origin:** LUAD typically originates in peripheral lung regions, whereas LUSC arises in central bronchi (WHO classification 2021).
- **Pathogenesis:** Atypical adenomatous hyperplasia and adenocarcinoma in situ — lesions originating from type II pneumocytes and/or club cells — are considered precursors of invasive LUAD (Park et al., 2018; Sivakumar et al., 2017). In contrast, LUSC develops from squamous dysplasia and squamous carcinoma in situ, which emerge from bronchial epithelial cells (WHO classification 2021).

- Histology and immunohistochemistry: Both subtypes show distinct cytomorphology, growth patterns and marker expression profiles (Rossi et al., 2013; Pelosi et al., 2013).
- Molecular profile: LUAD and LUSC differ markedly in their genomic landscapes. LUAD often harbors oncogenic driver mutations such as *EGFR*, *ALK*, or *KRAS*, which are less frequent in LUSC (Cancer Genome Atlas Research Network, 2012, 2014).
- Therapeutic implication: Targeted treatment options and response to immunotherapy differ substantially between the two subtypes, reflecting their divergent molecular and immunological profiles (Li et al., 2022).

Methodological rationale

We performed niche discovery separately for LUAD and LUSC to avoid assuming shared spatial biology, which might have limited the discovery of subtype-specific niche patterns. To still identify potential cross-subtype similarities, we aligned niches at the phenotype level using partial optimal transport to match niche pairs across subtypes (Figure 5A+B, Methods, p. 37, Supplementary Figure S10)

To address this important point, we now provide more detailed explanations for the separate analysis of LUAD and LUSC in the Methods section (p. 30).

[revised] “Rationale for separate analysis of LUAD and LUSC

We performed separate analyses for LUAD and LUSC based on clinicopathological and methodological rationales.

Clinicopathological rationale:

LUAD and LUSC are increasingly recognized as distinct tumor entities, based on well-established differences in etiology, histology, and molecular biology:

1. *Different etiology: In contrast to other types of lung cancer, LUAD also manifests in non-smokers, particularly in female non-smokers. In contrast, LUSC is strongly associated with smoking⁷⁷.*
2. *Different origin: LUSC typically arises from a main or lobar bronchus, with approximately two-thirds of cases occurring in the central lung compartment. In contrast, LUAD is more frequently observed in peripheral regions⁷⁸.*
3. *Different pathogenesis: The atypical adenomatous hyperplasia and adenocarcinoma in situ, which originate from type II pneumocytes and/or club cells, are regarded as the precursors of invasive LUAD^{79,80}. In contrast, squamous dysplasia and squamous carcinoma in situ are considered to be the precursors of LUSC, which arise in the bronchial epithelium⁷⁸.*
4. *Different histology: The histomorphology of LUAD and LUSC is markedly disparate, exhibiting distinct cytomorphology, growth patterns, and immunohistochemical expression profiles^{81–83}.*
5. *Different molecular landscape: LUAD and LUSC exhibit markedly disparate molecular signatures^{84,85}. For instance, several oncogenic driver gene alterations, such as *EGFR* or *ALK* have been identified in LUAD, which are typically not present in LUSC.*

6. *Different treatment effects: Some targeted agents developed for LUAD have been shown to be largely ineffective against LUSC. In addition, studies have shown that the survival benefit of IO therapy is histology-dependent⁸⁶.*

Methodological rationale

Joint niche discovery would have required assuming shared spatial biology between LUAD and LUSC, potentially obscuring subtype-specific patterns. To preserve biological resolution, we performed niche discovery separately. For comparability, we aligned niches post hoc at the phenotype level using partial optimal transport to identify matched niche pairs across subtypes (see also Methods, p. 37)."

2. In LUAD specifically, were there any inclusion/exclusion criteria? Were all growth patterns represented, including those without stromal invasion (lepidic, papillary, conventional micropapillary)? Was there any relationship between growth pattern and cell niches?- this needs to be explored as growth pattern is so determinative of the tumour microenvironment.

We thank the reviewer for this important question. We included all major histological subtypes of invasive non-mucinous adenocarcinoma as defined by the WHO classification of thoracic tumors (5th ed., 2021) – namely lepidic, acinar, papillary, micropapillary, and solid patterns – as well as invasive mucinous adenocarcinomas in the study. Rare LUAD variants, including minimally invasive adenocarcinoma, colloid adenocarcinoma, fetal adenocarcinoma, and enteric-type adenocarcinoma, were excluded due to low frequency and distinct biological characteristics. All diagnoses and growth pattern classifications were reviewed by two board-certified pathologists with over six (S.S.) and ten (F.K.) years of experience in thoracic pathology.

In response to the reviewer's suggestion, we conducted an additional analysis to explore potential associations between histological growth patterns and cell niche patterns. Since niche patterns directly correspond to the (cell niche-derived) risk scores, we compared the histological growth patterns with risk scores. This analysis yielded no statistically significant relationships supporting the idea that cell niches represent an independent aspect of lung cancer biology (**Rebuttal Figure 1**).

To illustrate these findings, we have added a new Supplementary Figure S17 (Rebuttal Figure 2), which compares growth patterns as well as other clinicopathological covariates across the cell niche-derived risk groups. The results are also referenced in the revised Results section. Case selection criteria and histopathological classification procedures are now described in greater detail in the revised Methods section and further illustrated in the new Supplementary Figure S24 (Rebuttal Figure 3).

Rebuttal Figure 1 (also part of new Supplementary Figure S17) : No relevant correlation between cell niche-derived risk groups and growth patterns.

(A) Histogram showing the distribution of LUAD growth patterns across cell niche-derived risk groups (RS1-3) which reveal no significant association (Chi-squared test).

(B) Histogram showing the distribution of LUSC growth patterns across cell niche-derived risk groups (RS1-3) which reveal no significant association (Chi-squared test).

Results section (p. 19):

[added] “Additionally, we compared risk groups with important clinical confounders⁶⁶ such as mutation status of frequently mutated genes in lung cancer as well as other known covariates, including sex, age, ECOG performance status, smoking status, tumor grade, thyroid transcription factor 1 (TTF-1) expression, and growth pattern, and found no relevant association (Supplementary Figures S16+17).”

Methods section (p. 30):

[added] “Diagnoses, histological growth patterns, tumor grading, pTNM classification, angioinvasion, lymphatic invasion, and tumor staging (according to the 8th edition of the TNM classification, AJCC) were reviewed by two board-certified pathologists with over six (S.S.) and ten (F.K.) years of experience in thoracic pathology. Histological subtyping was based on hematoxylin and eosin (H&E), Periodic Acid–Schiff (PAS), and Van Gieson staining. Routine diagnostic immunohistochemical panels used for tumor classification included p40, p63, CK5/6, CK7, and TTF-1 (Supplementary Figure S24). The diagnosis of LUAD was based on the presence of glandular growth patterns and typically showed CK7 positivity and p63/p40 as well as CK5/6 negativity, with or without co-expression of TTF-1. LUSC was diagnosed in the presence of keratinization and/or intercellular bridges in combination with positivity for p40, p63, and CK5/6 and no or only weak CK7 expression. All major histological subtypes⁷⁷ of invasive non-mucinous adenocarcinoma – namely lepidic, acinar, papillary, micropapillary, and solid patterns – as well as invasive mucinous adenocarcinomas were included in the study (see Supplementary Figure S24). Rare LUAD variants⁷⁷, including minimally invasive adenocarcinoma, colloid adenocarcinoma, fetal adenocarcinoma, and enteric-type adenocarcinoma, as well as large cell lung carcinomas were excluded due to low frequency and distinct biological characteristics. Neuroendocrine tumors – identified by positivity for synaptophysin and chromogranin A or CD56 – including small cell lung carcinoma (SCLC),

large cell neuroendocrine carcinoma (LCNEC), and typical or atypical carcinoids, were excluded from all analyses because they represent a separate tumor group as defined by the WHO.”

Rebuttal Figure 2 (also new Supplementary Figure S17): Cell-niche derived risk scores are independent of established clinicopathological covariates.

(A-G) Distribution of sex, age, ECOG performance status, smoking status, tumor grade, TTF-1 expression, and growth pattern across cell-niche-derived risk groups in LUAD tested for association using Chi-squared test.

(H-M) Distribution of sex, age, ECOG performance status, smoking status, tumor grade, and growth pattern across cell-niche-derived risk groups in LUSC tested for association using Chi-squared test.

Rebuttal Figure 3 (also new Supplementary Figure S24): Diagnostic workflow and case selection criteria for lung cancer specimens.

Schematic overview of the diagnostic work-up based on histomorphology and immunohistochemistry, including inclusion and exclusion criteria applied for LUAD and LUSC case selection. Rare tumor types and neuroendocrine neoplasms were excluded as outlined.

NSCLC, non-small cell lung cancer; SCLC, small-cell lung carcinoma; MIA, minimally invasive adenocarcinoma; ETAC, enteric-type adenocarcinoma; CA, colloid adenocarcinoma; FLAC, fetal adenocarcinoma; LCLC, large-cell lung carcinoma; LCNEC, large-cell neuroendocrine carcinoma; TTF-1, thyroid transcription factor 1; CK, cytokeratin; SYP, Synaptophysin; CgA, Chromogranin A; LUAD, lung adenocarcinoma; LUSC, lung squamous cell carcinoma.

3. Figures throughout would benefit from slightly better readability - larger text, inclusion of colour bars in heatmaps to define colour scales, annotations to highlight key results, axes labels and titles

We apologize for the limited figure readability in the original submission and thank the reviewer for pointing this out. In the revised manuscript, we have carefully revised all figures to enhance their clarity. Text size has been increased for all figures, color bars have been added to all heatmaps to clarify the data scales, and axis labels and figure titles have been adjusted for improved readability. In addition, we have added visual annotations to highlight key results, particularly in the heatmaps.

4. Smaller points moving through the text:

In the introduction, some more detail about the TME in NSCLC would better set the stage for the rest of the article

Could fig 1 be a supplementary figure? (From panel B onwards)

We thank the reviewer for these constructive suggestions.

In response to the first point, we have expanded the Introduction to describe the TME in NSCLC in greater detail. Specifically, we added the following paragraph to better highlight its prognostic relevance:

Introduction (p. 3):

[added] *“In NSCLC specifically, the TME has emerged as a key prognostic factor. For instance, Backman et al. recently demonstrated that the individual densities of lymphocyte subsets, the combined abundance of distinct immune cell populations, and their distances to one another each contribute independently to prognostic information regarding clinical outcomes²². More recent studies further support this view by showing that the spatial distribution of regulatory and cytotoxic T cells is closely associated with tumor stage and disease progression in NSCLC^{23,24}.”*

Regarding Figure 1, we agree that improved flow and focus in the main manuscript is an important goal. While we consider the figure helpful for introducing the analytical pipeline and giving readers a structured overview of the study design, we fully acknowledge that relocating it may improve clarity. We are therefore happy to move these panels to the Supplementary

Material if this step is supported by the editorial team and other reviewers. We appreciate the suggestion and are open to adapting the presentation accordingly.

Fig 2e, g what are the un-numbered groups?

For clarity, I'd recommend using different notation to refer to groups of cell niches (as in Fig 3k/4k) and groups of patients (as in Fig 3N/4N)

We thank the reviewer for pointing this out. The previously unnumbered groups in Figure 2e and 2g were indeed part of the defined clusters, but the labeling was not sufficiently clear in the original figure. This has now been corrected for clarity. In the case of LUSC, one cluster group (cluster group 7) was not explicitly labeled because it displayed a relatively unspecific cellular composition and did not clearly correlate with a defined immune context relevant for prognosis or therapeutic stratification. In the revised figure and legend, all clusters are now clearly annotated and described – including the previously unlabeled Cluster 7, which is now also discussed in the revised Results section, p. 14.

To further improve clarity and avoid confusion between cell niche groups and patient-level classifications, we have introduced distinct notations throughout the revised figures: niche groups (Fig. 3K/4K) are labeled using lowercase letters (a–e), while patient groups (Fig. 3N/4N) are indicated by Roman numerals (I–IV).

Results section (p. 14):

[added] “We also identified a seventh cluster in LUSC that was less specifically defined, showing low levels of immunosuppressive carcinoma cells (9.17/mm²) and moderate B cell (19.73/mm²) and T cell infiltration (260.56/mm²).”

In Fig 3n, Fig 4n, some cases are not assigned a group (no roman numeral on the heatmap). Why is that?

We thank the reviewer for this important observation. The unassigned cases in Figures 3n and 4n were in fact part of Cluster Group IV, but were mistakenly excluded due to an error in the visual boundary definition of the cluster. We have corrected this in the revised figures, and all cases are now properly assigned to their respective cluster groups.

Throughout page 14-17, there are many comparisons made between LUAD and LUSC but these are not backed by statistical testing

We apologize for not providing p-values and thank the reviewer for bringing it to our attention. In the revised version of the manuscript, we have systematically reviewed all comparative statements between LUAD and LUSC and have now added statistical analyses to support these comparisons.

How were the risk scores calculated and what was the rationale for 3 risk group categories?

We thank the reviewer for this question. Below, we outline our approach to calculating risk scores and explain the reasoning behind the choice of three risk groups. A more detailed justification for this stratification has also been added to the revised Methods section.

Risk score calculation: We thank the reviewer for this question. To calculate risk scores, we used the multivariate Cox regression model trained on the relative prevalence of different cell niche types. For each tissue spot, the model provides a continuous risk estimate. Since we use multiple tissue spots for each patient, we defined the patient's overall risk score by the highest risk value observed among their spots ("hot-spot" approach). This reflects the clinical rationale that the most aggressive tumor region is most relevant for prognosis, even if other areas appear less risky. For stratification into three groups (low, intermediate, high risk), we applied standard tertile cut-offs of the continuous risk scores. This choice balances interpretability and statistical power, while aligning with common practice in survival studies (for a more technical description, see Methods, p. 35).

Rationale of three risk groups:

Patients were stratified into low-, intermediate-, and high-risk groups by dividing the distribution of patient-level risk scores in the validation cohort into tertiles. This three-tier approach mirrors the UICC clinical staging system for early stage lung cancer (stages I-III), facilitating integration with existing workflows. It also provides more granularity than a simple low/high split while keeping the stratification actionable in both clinical decision-making and trial design. The use of tertiles for risk grouping is common across oncology domains (e.g., lung, prostate, breast) because it balances interpretability, statistical robustness, and clinical utility.

Methods section (p. 36):

[added] *"The use of three risk groups provides a natural counterpart to the UICC clinical staging system for early stage lung cancer (stages I-III), according to which patients in our study were classified. This alignment facilitates integration with established staging frameworks and enhances clinical interpretability. Stratifying patients into low, intermediate, and high-risk groups is also a widely used and practical approach across cancer types, including lung, prostate, and breast cancer. Compared to a simple low/high split, the three-tiered system offers more nuanced risk assessment while remaining actionable in routine care and clinical trial design."*

Reviewer #2

NCOMMS-25-40035

“AI-powered spatial cell phenomics enhances risk stratification in non-small cell lung cancer”

The authors study resected tumor specimens from 1,168 non-small cell lung cancer (NSCLC) patients, divided between lung adenocarcinoma (LUAD) and lung squamous carcinomas (LUSC) that came from two independent German cancer centers (Berlin and Cologne). They obtain H&E stained images of the tumors and also images of immunohistochemical stains (IHC) for 12 immune cell related markers and developed computational biology “AI-powered spatial cellomics approach”) approaches to analyze these image data sets and classify tumors, and the tumor microenvironment including the identification of “niches”. In addition, they determined the spatial relationship of tumor cells to the TME niches. All of these image data were computationally sorted into groups (“clusters”) and also related to survival data. Their image analyses improved survival annotation by some 14% for LUAD and 47% for LUSCs. They conclude: “ Our results showed that complex immune cell niche patterns identify potentially undertreated high-risk patients qualifying for adjuvant therapy. Our approach highlights the potential of AI-powered multiplex imaging analyses to better understand the contribution of the TME to cancer progression and to improve risk stratification and treatment selection in NSCLC. “

Comments to the authors:

This manuscript is reviewed in the setting of the urgent need to use image analysis and computational biology approaches on routine pathologic materials, as well as IHC stains, to classify lung tumors in a manner that would enhance the selection of the best therapy and obtain prognostic information for individual patients. In addition, to be able to independently validate the authors’ approaches and results, compare these to multiple other approaches, and to be able to use the large datasets for further research outside of the authors’ domains.

1. The amount of work is very large, clearly presented, and appears to be carefully done. The authors are to be commended on the scale and detailed analyses they have performed.

We thank the reviewer for the positive feedback.

2. The ability of other investigators to obtain their image and clinical datasets easily for both independent validation, and to use for comparison to other datasets is very important. I believe that is the authors’ intent as well, but it will be important for this to be clearly stated and the conditions and methods for obtaining these key data spelled out. With such access the value and reason for publishing this paper is greatly enhanced.

To facilitate independent validation and enable direct comparison with other datasets and promote further research, all processed cell-classification results and niche annotations will be made publicly available on Zenodo. Due to file size limits, a second Zenodo repository will be

used for high-resolution image tiles for each patient, including all multiplex immunofluorescence (mIF) channels and H&E, totaling 14,000 images.

<https://zenodo.org/records/11395885> + <https://zenodo.org/records/16882468>

We have added a statement on data availability including the repository links in the Data and software availability section (p. 39).

[added] *“All processed cell-classification results and niche annotations are publicly available on Zenodo (DOI: 10.5281/zenodo.11395885). High-resolution image tiles for each patient, including all multiplex immunofluorescence (mIF) channels and H&E (14,000 images), are provided in a second Zenodo repository (DOI: 10.5281/zenodo.16882468). The complete raw dataset (>1 TB), including patient metadata and full image files, is available via a secure S3 bucket upon request to the corresponding author. Access requires only a signed non-commercial use declaration and an academic email address; all bona fide academic requests will be granted. Code is available at https://github.com/gabrieldernbach/cell_niches.*

3. There have been a large number of other publications dealing with various computational approaches to image analyses of lung cancers and other tumor types. While it is impractical to try to compare their bespoke approach to a large number of these, it would be useful to know if the authors took a few of the most prominent other publications dealing with identification of the tumor microenvironment from H&E images and compared them to their own approach. Are the various groups and niches identified by the different approaches similar or quite distinct? Is there any value of combining the approaches and subsequent classifications the authors developed with classes identified by other approaches?

We thank the reviewer for raising this important point. Several recent high-profile studies have characterized the TME in NSCLC using either H&E-based deep learning or multiplexed immunohistochemistry. These include analyses focused primarily on tertiary lymphoid structures (Chen et al., 2024. Deep learning on tertiary lymphoid structures in hematoxylin-eosin predicts cancer prognosis and immunotherapy response. NPJ Precision Oncology), LUAD-specific cellular neighbourhoods (Sorin et al. 2023. Single-cell spatial landscapes of the lung tumour immune microenvironment. Nature), spatially resolved tumor-immune microenvironments in smaller cohorts of 51 LUAD and 51 LUSC patients (Desharnais et al., 2025. Spatially mapping the tumour immune microenvironments of non-small cell lung cancer. Nature Commun.), and large-scale spatial immunophenotyping in both tumor types, focusing on cell distances between cell type pairs (Backman et al., Spatial immunophenotyping of the tumour microenvironment in non-small cell lung cancer., 2023. European Journal of Cancer). In addition, Cords et al., 2024 (Cancer-associated fibroblast phenotypes are associated with patient outcome in non-small cell lung cancer) concentrated on the cell population of cancer-associated fibroblasts, linking distinct phenotypes to patient prognosis.

Across these studies, we observe both notable **similarities** and important **differences** compared with the spatially resolved cell niches and niche patterns identified in our work.

Similarities exist with respect to cellular niches in both LUAD and LUSC that showed similar compositions with those reported in previous studies. Similar to the LUAD-specific analysis by Sorin et al., 2023, we identified two B cell-rich niches (LUAD niches 5 and 6) accompanied by high frequencies of regulatory T cells, which were associated with worse prognosis. In our dataset, these niches were enriched in higher-risk groups RS2 and RS3 compared with the low-risk group RS1. Likewise, our TLS-rich niche in LUSC (LUSC niche 7) was also associated with improved prognosis, as shown by Chen et al., 2024, and was markedly enriched in RS1 compared with the high-risk groups RS2 and RS3. We also confirmed the poor-prognosis association of macrophage-rich neighbourhoods in both LUAD and LUSC, as reported by Backman et al., 2023, and by Desharnais et al., 2025 for LUAD only. The macrophage-rich, cold LUAD niche 10 we observed was more prevalent in the higher-risk groups, as were the macrophage-rich LUSC niches 9 and 10.

Differences were observed with respect to the following points. First, our histology-stratified analysis revealed divergent prognostic associations between specific niches that were not apparent when LUAD and LUSC were analysed together. In contrast to our findings similar to those reported by Sorin et al., 2023 for LUAD, our analyses differed for LUSC. We identified two B cell- and regulatory T cell-rich niches (LUSC niches 5 and 7) enriched in the low-risk group RS1, whereas a third niche (LUSC niche 8) was enriched in the higher-risk groups RS2 and RS3. The TLS-rich niche was associated with a favorable prognosis in LUSC (LUSC niche 7), but occurred more frequently in higher-risk groups in LUAD (LUAD niche 5), whereas Chen et al., 2024 reported a favorable association in NSCLC independent of the histological subtype. Notably, their association lost statistical significance when adjusted for the UICC stage, in contrast to the results from our approach. Macrophage-rich niches were linked to poor prognosis in both histologies in our data (LUAD niche 10; LUSC niches 9 and 10), whereas previous reports, such as Desharnais et al., 2025, found a favourable association in LUSC. In our analyses, we also observed a macrophage-rich cold niche in LUAD that, in contrast to other studies, was particularly enriched in low-risk patients (LUAD niche 1). Furthermore, we identified additional niches not described in these previous studies, namely the immune-cold niches LUAD 3 and LUAD 7, as well as LUSC 4 and LUSC 6, which were associated with extremely poor patient prognosis and were particularly enriched in risk group 3. These niches were characterized by a high density of tumor cells or immunosuppressive tumor cells. Notably, in LUSC, the immunosuppressive tumor niche 6, when occurring in combination with the TLS-rich niche 7, was associated with improved survival (risk group 1), further underscoring the importance of considering niche patterns as applied in our approach rather than isolated niches for prognostic prediction.

As an additional perspective, recent work by Cords et al., 2024 linked distinct cancer-associated fibroblast phenotypes to patient prognosis in NSCLC. Since fibroblasts were not included as a cell population in our initial analysis, examining the relationship between fibroblast abundance and niche composition could be important to consider the prognostic effects of fibroblasts. To explore this, we additionally performed fibroblast quantification from H&E images and analysed their distribution across niches and prognostic associations, finding that fibroblast abundance showed weak to moderate correlations with cell niches. Furthermore, fibroblasts were significantly enriched in “hot” niches compared to “cold” immunosuppressive niches in both

LUAD and LUSC, but showed no statistically significant association with patient survival when patients were stratified by fibroblast abundance. A limitation of our approach is that, unlike Cords et al., we could not distinguish fibroblast subpopulations due to the absence of the mIF markers required for subtype definition, preventing a detailed comparison with their analyses; incorporating such fibroblast subtyping into niche pattern analysis would be a promising avenue for future studies.

We added these additional results and considerations to the results and methods section and the discussion in the revised manuscript including new Supplementary Figures S18 and S19 (Rebuttal Figures 1+2).

Results section (p. 19):

[added] *“LUAD with niche risk score 1 (RS1) showed a predominant mixture of hot niches (Leuco^{high}, Infl^{high}, T cell^{high}), which is consistent with the literature. Several studies have shown that high densities of tumor-infiltrating lymphocytes (TILs) – such as CD3+TILs, CD4+TILs, CD8+TILs and CD20+TILs – are favorable prognostic biomarkers^{17,34,67–69}. In line with the LUAD-specific analysis by Sorin et al., 2023, two B cell-rich niches with high regulatory T cell content (TLS^{high}, Lymph^{high I}) were also enriched in our high-risk groups RS2 and RS3, further supporting their association with poor prognosis in LUAD. In addition, a higher concentration of the cold TAM-dominated niche (LUAD Mφ^{high}) was observed at the higher risk scores. This is consistent with the observation that an increased incidence of tumor-associated macrophages (TAMs) is associated with reduced overall survival in various cancer types, and aligns with reports by Backman et al., 2023 and Desharnais et al., 2025 (LUAD) linking macrophage-rich niches to poor prognosis.^{58,70} We also identified additional cold immune niches (Immu^{NULL}, Immunosup^{high}, and Leuco^{low}) characterized by high densities of tumor or immunosuppressive tumor cells, which were strongly enriched in RS3 and associated with particularly poor prognosis – patterns not reported in the previously cited large-scale spatial studies. Similar to LUAD, for LUSC higher concentrations of hot niches (TLS^{high}, LUSC Infl^{high}) were found at risk score 1, whereas the cold niches Leuco^{low} and Immu^{NULL} accumulated at risk scores 2 and 3. Again, cold macrophage niches (Mφ^{high I}; Mφ^{high II}) were enriched in high-risk tumors. In contrast to LUAD, TLS^{high} niche and Immunosup^{high} niche were both particularly observed in risk score 1 tumors, whereas Chen et al., 2024 reported a favorable association of TLS enrichment in NSCLC without histology-specific separation. This underlined the distinct nature of LUAD and LUSC. We also observed that the B cell- and regulatory T cell-rich niche Lymph^{high I} was enriched in RS1, whereas a second such niche (Lymph^{high II}) was enriched in higher-risk groups, emphasizing entity-specific prognostic associations and highlighting the importance of considering niche patterns rather than single niches for prognostic prediction. In a complementary analysis, we quantified fibroblast abundance from H&E images and analysed their distribution across the cell niches. Fibroblast abundance showed weak to moderate correlations with niche frequencies and was significantly higher in “hot” niches compared to “cold” niches in both LUAD and LUSC, but did not show a statistically significant association with patient survival (Supplementary Figures S18+19).”*

Discussion section (p. 29):

[added] *“In this context, cancer-associated fibroblasts (CAFs) represent a particularly relevant stromal cell population, as recent work by Cords et al., 2024 demonstrated that distinct CAF phenotypes are linked to either favorable or poor prognosis in NSCLC. While our current analysis did not differentiate CAF subtypes due to the absence of specific mIF markers, incorporating such phenotypic distinctions in future studies could further improve prognostic stratification and expand our understanding of stromal–immune–tumor interactions.”*

Methods section (p. 39):

[added] “Association of risk scores with fibroblast abundance

Fibroblasts were detected in H&E-stained images using the RudolfV¹⁰¹ pathology foundation model. The model was applied to all tissue spots in the LUAD and LUSC cohorts. For each patient, fibroblast abundance was quantified per spatial niche both as absolute counts and as proportions relative to all cells within the niche. Correlations between fibroblast abundance and niche frequencies were computed using Pearson’s correlation coefficient. Differences in fibroblast abundance between “hot” and “cold” niches were assessed using a Chi-square test. To evaluate prognostic associations, patients were stratified into tertiles based on either their fibroblast maximum abundance over tissue spots (max pooling) or their fibroblast mean abundance over tissue spots, analogous to the risk score analyses for niche patterns. Survival differences between strata were analysed using Kaplan–Meier curves and log-rank tests.”

Fibroblast Other cell

Rebuttal Figure 1 (also new Supplementary Figure S18): Fibroblast abundance across cell niches in LUAD.

- (A) Representative H&E image of a tissue spot containing LUAD (left) and corresponding high-magnification zoom-in (right).
- (B) Output of the cell classification model showing AI-derived fibroblast predictions (red polygons) and other cell predictions (blue polygons) for the same spot and corresponding zoomed region shown in (A).
- (C) Correlation of fibroblast abundance with niche abundance across tissue spots; each dot represents one tissue spot.
- (D) Absolute fibroblast counts per cell niche.
- (E) Relative fibroblast frequency per cell niche, with niches classified as *hot* (red bar at bottom) or *cold* (blue bar at bottom).
- (F) Relative fibroblast frequency within cold (left) and hot (right) niche groups.
- (G) Kaplan-Meier analysis stratified by fibroblast maximum abundance over tissue spots per patient (max pooling); patients were divided into tertiles for direct comparison with cell niche pattern-based risk scores.
- (H) Kaplan-Meier analysis stratified by fibroblast mean abundance over tissue spots per patient; patients were divided into tertiles for direct comparison with cell niche pattern-based risk scores.

Fibroblast Other cell

Rebuttal Figure 2 (also new Supplementary Figure S19): Fibroblast abundance across cell niches in LUSC.

- (A) Representative H&E image of a tissue spot containing LUSC (left) and corresponding high-magnification zoom-in (right).
- (B) Output of the cell classification model showing AI-derived fibroblast predictions (red polygons) and other cell predictions (blue polygons) for the same spot and corresponding zoomed region shown in (A).
- (C) Correlation of fibroblast abundance with niche abundance across tissue spots; each dot represents one tissue spot.
- (D) Absolute fibroblast counts per cell niche.
- (E) Relative fibroblast frequency per cell niche, with niches classified as *hot* (red bar at bottom) or *cold* (blue bar at bottom).
- (F) Relative fibroblast frequency within cold (left) and hot (right) niche groups.
- (G) Kaplan–Meier analysis stratified by fibroblast maximum abundance over tissue spots per patient (max pooling); patients were divided into tertiles for direct comparison with cell niche pattern-based risk scores.
- (H) Kaplan–Meier analysis stratified by fibroblast mean abundance over tissue spots per patient; patients were divided into tertiles for direct comparison with cell niche pattern-based risk scores.

4. A key aspect of course would be to integrate current widely used in standard clinical pathology IHC stain assessments of NSCLCs with the authors' image analyses such as TTF-1, Napsin A, CK5/6, p63, p40, synaptophysin, chromogranin A, CD56. It would seem likely that information on some of these routinely used IHC markers for NSCLC diagnosis are available and if so, it would be very helpful to see how this information integrated with the authors' image analyses classes and niches.

We thank the reviewer for raising this point. The IHC stains mentioned by the reviewers were indeed used to classify the tumors as part of the diagnostic work-up in accordance with the WHO Classification of Thoracic Tumors (5th ed., 2021) and therefore served as a basis for the subsequent niche-pattern analysis. Diagnoses based on histomorphological features were supported by immunohistochemical panels including TTF-1, CK7, p40, p63, and CK5/6. LUAD was diagnosed based on the presence of glandular growth patterns and typically showed CK7 positivity and p40/p63 as well as CK5/6 negativity, with or without co-expression of TTF-1. In contrast, LUSC was defined by keratinization and/or intercellular bridges in combination with positivity for p40, p63, and CK5/6 and no or only weak CK7 expression. Neuroendocrine tumors such as small cell lung carcinomas, large cell neuroendocrine carcinomas, and carcinoids – which are characterized by positivity for synaptophysin and chromogranin A as well as CD56 – were excluded from all analyses as they represent a separate tumor group with distinct biology according to the WHO classification. These marker profiles were essential in assigning each tumor to its respective histological subtype and were therefore already fully integrated prior to niche analysis. As such, these IHC markers do not represent independent variables in relation to the niche structure but rather define the biological classification based on which downstream analysis was performed – with the exception of TTF-1, which may be positive or negative in LUAD carcinoma cells and whose expression status is a known prognostic marker. We therefore specifically analyzed the TTF-1 expression in relation to the TME-based risk groups and found no significant associations. These results are described in the revised Results section and visualized in the new Supplementary Figure S17.

To illustrate the diagnostic procedure, including the role of immunohistochemistry, we have added a new Supplementary Figure S25 (Rebuttal Figure 3). Case selection criteria and histopathological classification procedures are now described in greater detail in the revised Methods section.

Methods section (p. 30):

[added] *“Diagnoses, histological growth patterns, tumor grading, pTNM classification, angioinvasion, lymphatic invasion, and tumor staging (according to the 8th edition of the TNM classification, AJCC) were reviewed by two board-certified pathologists with over six (S.S.) and ten (F.K.) years of experience in thoracic pathology. Histological subtyping was based on hematoxylin and eosin (H&E), Periodic Acid–Schiff (PAS), and Van Gieson staining. Routine diagnostic immunohistochemical panels used for tumor classification included p40, p63, CK5/6, CK7, and TTF-1 (Supplementary Figure S24). The diagnosis of LUAD was based on the presence of glandular growth patterns and typically showed CK7 positivity and p63/p40 as well as CK5/6 negativity, with or without co-expression of TTF-1. LUSC was diagnosed in the presence of keratinization and/or intercellular bridges in combination with positivity for p40, p63, and CK5/6 and no or only weak CK7 expression. All major histological subtypes⁷⁷ of invasive non-mucinous adenocarcinoma – namely lepidic, acinar, papillary, micropapillary, and solid patterns – as well as invasive mucinous adenocarcinomas were included in the study (see Supplementary Figure S24). Rare LUAD variants⁷⁷, including minimally invasive adenocarcinoma, colloid adenocarcinoma, fetal adenocarcinoma, and enteric-type adenocarcinoma, as well as large cell lung carcinomas were excluded due to low frequency and distinct biological characteristics. Neuroendocrine tumors – identified by positivity for synaptophysin and chromogranin A or CD56 – including small cell lung carcinoma (SCLC), large cell neuroendocrine carcinoma (LCNEC), and typical or atypical carcinoids, were excluded from all analyses because they represent a separate tumor group as defined by the WHO.”*

Results section (p. 19):

[added] *“Additionally, we compared risk groups with important clinical confounders⁶⁶ such as mutation status of frequently mutated genes in lung cancer as well as other known covariates, including sex, age, ECOG performance status, smoking status, tumor grade, thyroid transcription factor 1 (TTF-1) expression, and growth pattern, and found no relevant association (Supplementary Figures S16+17).”*

Rebuttal Figure 3 (also new Supplementary Figure S24): Diagnostic workflow and case selection criteria for lung cancer specimens.

Schematic overview of the diagnostic work-up based on histomorphology and immunohistochemistry, including inclusion and exclusion criteria applied for LUAD and LUSC case selection. Rare tumor types and neuroendocrine neoplasms were excluded as outlined.

NSCLC, non-small cell lung carcinoma; SCLC, small-cell lung carcinoma; MIA, minimally invasive adenocarcinoma; ETAC, enteric-type adenocarcinoma; CA, colloid adenocarcinoma; FLAC, fetal adenocarcinoma; LCLC, large-cell lung carcinoma; LCNEC, large-cell neuroendocrine carcinoma; TTF-1, thyroid transcription factor 1; CK, cytokeratin; SYP, Synaptophysin; CgA, Chromogranin A; LUAD, lung adenocarcinoma; LUSC, lung squamous cell carcinoma.

5. Likewise, my guess is that many of these cases had routine oncogenotypes obtained as part of standard of care, such as EGFR, KRAS, TP53, STK11/LKB1 and others. If so it would be helpful to see how these are integrated and compare to the authors' classifications.

We thank the reviewer for this important comment. To address this point, we analyzed targeted panel sequencing data (covering 18 genes) from a substantial subset of patients (n=325) in our cohort (see new Supplementary Table S12 for details). In both LUAD and LUSC, no statistically significant associations were found between TME-based risk groups and the mutational status of frequently altered genes. The results have been added as new Supplementary Figure S16 (Rebuttal Figure 4) and are now described in the revised Results section.

A

Gene	BRAF	CTNNB1	DDR2	EGFR	ERBB2	FGFR1	FGFR2	FGFR3	IDH1	KEAP1	KRAS	MAP2K1	MET	NRAS	PIK3CA	PTEN	STK11	TP53
Exon	11, 15	3	3-18	18-21	19,20	4-7, 10, 12-15	8-10, 12, 17, 20	7, 10, 15	4	2-6	2,3	2	14	2, 3	10, 21	1-8	2-7	5-8

Rebuttal Figure 4 (also new Supplementary Figure S16): Independence of cell-niche derived risk groups and mutation status.

(A) Gene panel overview, including 18 essential genes for the molecular characterization of NSCLC.
 (B) Mutation frequencies across cell-niche-derived risk groups in LUAD tested for association using Chi-squared test.
 (C) Mutation frequencies across cell-niche-derived risk groups in LUSC tested for association using Chi-squared test.

Results section (p. 19):

[added] “Additionally, we compared risk groups with important clinical confounders⁶⁶ such as mutation status of frequently mutated genes in lung cancer as well as other known covariates, including sex, age, ECOG performance status, smoking status, tumor grade, thyroid transcription factor 1 (TTF-1) expression, and growth pattern, and found no relevant association (Supplementary Figures S16+17).”

6. I did not find a clear statement about analysis of some standard clinical demographic data with the classifications including age, gender, smoking status, race. I presume these analyses were done and if they did not impact on the classification (or if they did) this should be stated. Likewise as part of the deposited datasets available for other investigators to use, it will be important to also have some basic clinical demographic data such as these available as well.

We thank the reviewer for this important comment. We analyzed clinical-demographic variables – including sex, age, ECOG performance status, smoking status, and other available covariates – in relation to the TME-based risk groups and found no significant associations. These findings are mentioned in the Results section and supported by the new Supplementary Figure S17 (Rebuttal Figure 5). As suggested, we also provide a comprehensive overview of clinico-pathological characteristics in the new Supplementary Table S1. We would like to note that race/ethnicity is not routinely collected in clinical records in most European healthcare systems, including the institutions contributing to this study. Consequently, this variable was not available for analysis.

Rebuttal Figure 5 (also new Supplementary Figure S17): No relevant correlation between cell-niche derived risk groups and established clinicopathological covariates.

(A-G) Distribution of sex, age, ECOG performance status, smoking status, tumor grade, TTF-1 expression, and growth pattern across cell-niche-derived risk groups in LUAD (Chi-squared test).

(H-M) Distribution of sex, age, ECOG performance status, smoking status, tumor grade,, and growth pattern across cell-niche-derived risk groups in LUSC (Chi-squared test).

Results section (p. 19):

[added] *“Additionally, we compared risk groups with important clinical confounders⁶⁶ such as mutation status of frequently mutated genes in lung cancer as well as other known covariates, including sex, age, ECOG performance status, smoking status, tumor grade, thyroid transcription factor 1 (TTF-1) expression, and growth pattern, and found no relevant association (Supplementary Figures S16+17).”*

Reviewer #3

The authors use two large cohorts for training and testing of a model that predicts the clinical outcomes of lung cancer patients. Adeno- and squamous lung cancers are analyzed separately for 10 cell niches with a circle with a diameter of 34 microns, i.e. the size of a macrophage, in the TME. The niches are divided into hot and cold immune cell regions based on lymphocyte content using 12 antibodies in a cyclic IF staining protocol followed by an H&E stain. Adding the niches to the TNM classification improves the survival prediction. Most importantly, the niches identify TNM stage-I cases who would benefit from adjuvant chemotherapy. Altogether, adding niche information increases the c-index above the TNM UICC8 baseline.

Workflow: pixel level registered 12-plex IF and H&E tiles from TMA blocks are used to train a risk score predictor. Each case is represented by 4 cores in the TMA.

The pipeline consists of the following steps:

Stage 1: tissue segmentation into cancer, stroma, benign, necrosis. A U-Net model trained in house possesses an average F1-score = 0.92

Stage 2: cell identification. Nuclear segmentation was performed with an optimized StarDist model, providing an F1-score = 0.91

Stage 3: cell classification. For cell classification 12 independent ConvNet models were trained, using each mIF channel and the corresponding H&E as the input. The patch level F1-score = 0.91)

Stage 4: niche identification (K-mean clustering of the 34 μ m diameter neighborhoods for each cell; k is set to 10)

Stage 5: survival analysis: the niche composition of each core is used to predict patient survival using a multiple instance learning model. The highest risk core is used together with clinical variables is used to train a Cox regression model in the training cohort. The model is applied to the test cohort.

- Will the work be of significance to the field and related fields? How does it compare to the established literature? If the work is not original, please provide relevant references.

1. This work is significant to the field and related fields with regards to the analysis of the tumor microenvironment. This interesting study uses a large lung cancer cohort with high quality data. The main contribution to the field is that the analysis pipeline is straight forward and easy to recreate at a new local site.

2. Adding the results from the niche analysis to the TNM staging has the potential to improve the stratification for adjuvant therapy.

3. multiple consideration related to data structure and quality are addressed using the appropriate statistical methods, leading to rigorous data analysis.

4. The cell types depicted in ST2 demonstrate how the 12-plex is used to derive almost 200 different subpopulations of immune cells, which leads to the cluster composition and cell composition of clusters shown in the supplementary tables. This data is a significant contribution to the field.

- Does the work support the conclusions and claims, or is additional evidence needed? The conclusions are supported by the data.

- Are there any flaws in the data analysis, interpretation and conclusions? - Do these prohibit publication or require revision?

The major concerns are:

1. The immunosuppressive cancer phenotype could be caused by cells in the stroma, for example fibroblasts. Fibroblasts are not considered in the current study. Can the H&E images be used to determine the amount of fibroblasts in the TME and a relationship between numbers of fibroblasts and niche composition. Please consider the effects of fibroblasts as described in “Cords, L., Engler, S., Haberecker, M., Rüschoff, J.H., Moch, H., de Souza, N., and Bodenmiller, B. (2024). Cancer-associated fibroblast phenotypes are associated with patient outcome in non-small cell lung cancer. *Cancer Cell*” in the discussion of the paper.

We thank the reviewer for the suggestion to evaluate the relevance of fibroblast abundance. To address the potential role of fibroblasts in our dataset, we used a model based on RudolfV, a recent state-of-the-art pathology foundation model, to classify cells in H&E-stained images as fibroblast or other cell. Correlation analysis between fibroblast counts and niche frequencies across all tissue spots revealed mild to moderate correlations ($r = -0.28$ to 0.53 for LUAD; $r = -0.22$ to 0.53 for LUSC). We then quantified fibroblast abundance within each cell niche, both in absolute numbers and as relative proportions. Fibroblasts were significantly enriched in “hot” niches compared to “cold”, immunosuppressive niches in both LUAD and LUSC ($p < 0.0001^{****}$). We further assessed potential prognostic associations by stratifying patients according to fibroblast abundance using both the maximum abundance per patient and the mean abundance per patient, as applied for the niche patterns. Neither analysis yielded statistically significant associations with survival ($p = 0.315$; $p = 0.336$ for LUAD; $p = 0.114$; $p = 0.233$ for LUSC).

These findings indicate that fibroblasts are indeed associated with certain niche patterns, but also demonstrate that our niche patterns retain prognostic value independently of fibroblast abundance. A limitation of our approach is that, unlike Cords et al., 2024, we could not differentiate fibroblast subpopulations into good- and poor-prognostic cancer-associated fibroblasts because these markers were not included in our mIF panel. We included the fibroblast analyses into the revised Results, Discussion, and Methods sections, accompanied by new Supplementary Figures S18 (Rebuttal Figure 1) and S19 (Rebuttal Figure 2). We also added a paragraph in the discussion to address this point.

Results section (p.19):

[added] “In a complementary analysis, we quantified fibroblast abundance from H&E images and analysed their distribution across the cell niches. Fibroblast abundance showed mild to

moderate correlations with niche frequencies and was significantly higher in “hot” niches compared to “cold” niches in both LUAD and LUSC, but did not show a statistically significant association with patient survival (Supplementary Figures S18+19).”

Discussion section (p. 29):

[added] *“In this context, cancer-associated fibroblasts (CAFs) represent a particularly relevant stromal cell population, as recent work by Cords et al., 2024 demonstrated that distinct CAF phenotypes are linked to either favorable or poor prognosis in NSCLC. While our current analysis did not show any significant association between the abundance/density of fibroblasts in cell niches with risk scores, we were not able to take into account different CAF subtypes because fibroblast subtype markers were not included in our mIF panel, which we designed to limit complexity for easier clinical applicability. Future studies will have to evaluate the additional clinical value of incorporating such phenotypic distinctions.”*

Methods section (p. 39):

[added] **“Association of risk scores with fibroblast abundance**

Fibroblasts were detected in H&E-stained images using the RudolfV¹⁰¹ pathology foundation model. The model was applied to all tissue spots in the LUAD and LUSC cohorts. For each patient, fibroblast abundance was quantified per spatial niche both as absolute counts and as proportions relative to all cells within the niche. Correlations between fibroblast abundance and niche frequencies were computed using Pearson’s correlation coefficient. Differences in fibroblast abundance between “hot” and “cold” niches were assessed using a Chi-squared test. To evaluate prognostic associations, patients were stratified into tertiles based on either their fibroblast maximum abundance over tissue spots (max pooling) or their fibroblast mean abundance over tissue spots, analogous to the risk score analyses for niche patterns. Survival differences between strata were analysed using Kaplan–Meier curves and log-rank tests.”

Rebuttal Figure 1 (also new Supplementary Figure S18): Fibroblast abundance across cell niches in LUAD.

- (A) Representative H&E image of a tissue spot containing LUAD (left) and corresponding high-magnification zoom-in (right).
- (B) Output of the cell classification model showing AI-derived fibroblast predictions (red polygons) and other cell predictions (blue polygons) for the same spot and corresponding zoomed region shown in (A).
- (C) Correlation of fibroblast abundance with niche abundance across tissue spots; each dot represents one tissue spot.
- (D) Absolute fibroblast counts per cell niche.
- (E) Relative fibroblast frequency per cell niche, with niches classified as *hot* (red bar at bottom) or *cold* (blue bar at bottom).
- (F) Relative fibroblast frequency within cold (left) and hot (right) niche groups.
- (G) Kaplan-Meier analysis stratified by fibroblast maximum abundance over tissue spots per patient (max pooling); patients were divided into tertiles for direct comparison with cell niche pattern-based risk scores.
- (H) Kaplan-Meier analysis stratified by fibroblast mean abundance over tissue spots per patient; patients were divided into tertiles for direct comparison with cell niche pattern-based risk scores.

Rebuttal Figure 2 (also new Supplementary Figure S19): Fibroblast abundance across cell niches in LUSC.

- (A) Representative H&E image of a tissue spot containing LUSC (left) and corresponding high-magnification zoom-in (right).
- (B) Output of the cell classification model showing AI-derived fibroblast predictions (red polygons) and other cell predictions (blue polygons) for the same spot and corresponding zoomed region shown in (A).
- (C) Correlation of fibroblast abundance with niche abundance across tissue spots; each dot represents one tissue spot.
- (D) Absolute fibroblast counts per cell niche.
- (E) Relative fibroblast frequency per cell niche, with niches classified as *hot* (red bar at bottom) or *cold* (blue bar at bottom).
- (F) Relative fibroblast frequency within cold (left) and hot (right) niche groups.
- (G) Kaplan-Meier analysis stratified by fibroblast maximum abundance over tissue spots per patient (max pooling); patients were divided into tertiles for direct comparison with cell niche pattern-based risk scores.
- (H) Kaplan-Meier analysis stratified by fibroblast mean abundance over tissue spots per patient; patients were divided into tertiles for direct comparison with cell niche pattern-based risk scores.

2. The risk score stratification using niche variables improves the baseline c-index calculated based on UICC8 TNM variables only by a small amount in terms of c-index (proportion of patient pairs in which the one that has higher risk score experienced death sooner than the lower score one). However, the improvement compared to the baseline TNM UICC8 is small: e.g. the baseline c-index for LUSC is 0.630; and the model achieved a c-index of 0.692. It is unclear how the authors calculated the 47% improvement in the c-index that they claim.

We agree that the absolute difference in c-index values between the models appears modest at first glance. However, our statement of a 47% improvement is correct when interpreted relative to the informative range of the c-index, which lies between 0.5 (random prediction) and 1.0 (perfect prediction). Therefore, we calculated the relative improvement over the baseline model by comparing how far each model moves beyond the random baseline:

$$\text{Relative improvement} = \frac{0.692-0.5}{0.630-0.5} = \frac{0.192}{0.130} \approx 1.47$$

This corresponds to a 47% improvement of our niche-based model compared to the UICC8-based baseline in LUSC. We have added a detailed explanation of this rationale to the revised Methods section.

Methods section (p. 36):

[added] “To assess the improvement in outcome prediction of the cell niche model compared to the clinical baseline (UICC8), we computed the relative improvement in c-index, accounting for its informative range between 0.5 (random prediction) and 1.0 (perfect prediction). It is defined as follows:

$$\text{Relative improvement} = \frac{c-\text{model} - 0.5}{c-\text{baseline} - 0.5} ,$$

where c-model denotes the c-index of the cell niche model, c-baseline is the c-index of the clinical baseline (UICC8) and 0.5 represents the lower bound of the c-index, corresponding to random prediction.”

3. lack of clarity of several steps in the risk score determination:

a. it is unclear how the ConvNet prediction of 256 x 256 tiles will be processed to arrive at the niche classification.

The process works as follows: For each detected cell in the multiplex immunofluorescence (mIF) images, a 256 × 256 pixel tile centered on the cell is extracted and analyzed across all image channels. The ConvNet outputs binary predictions (positive/negative) for each antigen marker, resulting in a multi-hot vector that summarizes the marker expression profile of the cell.

To capture the spatial context, a 34 μm radius around each cell – corresponding to our predefined niche size – is considered. Within this neighborhood, the frequency of marker-positive cells is computed for each antigen. These local marker count vectors are normalized to account for variations in cell density.

To assign each neighborhood to a niche type, we used a predefined set of prototype niche profiles obtained from K-means clustering across the full dataset. Each cell’s local marker composition is then compared to these prototypes and assigned to the nearest cluster using a nearest-neighbour approach within the K-means embedding space. This results in a robust niche classification for each cell based on its microenvironmental context.

We have clarified this entire procedure in the revised manuscript.

Results section (p. 13):

[revised] *“Finally, the cell classification task was modeled as a multi-label classification task with twelve independent models – one per mIF channel – using a ConvNext³⁰ architecture. We chose independent models per mIF channel to counteract “Clever Hans” effects³¹, which we observed when using joint models that exploited marker correlations among the mIF channels. The cell classification models were applied to a total of 53 million cells and achieved an average F1-score of 0.91 on independent test cases (Figure 1D). The resulting binary predictions were aggregated into a multi-hot vector per cell, which formed the basis for downstream niche classification. As outlined in Figure 1E, niche composition was derived by comparing the local marker frequencies (within a 34 μm radius) to prototype distributions obtained via clustering. Full implementation details are provided in the Methods section.”*

Methods section (p. 34):

[revised] *“Using the predefined radius of 34 μm, we used a scipy KDTree⁹¹ to calculate the number of cells positive for each marker within this spatial neighborhood (see Methods section: Cell classification, page 33). These marker counts were logarithmically transformed to put more weight on the presence or absence of cells by gradually saturating the contribution increasing count, to further put an emphasis on the relative composition of a neighbourhood the expression*

values were divided by the local mass (percentage). To identify unique cellular neighborhoods, we performed Mini Batch K Means++ clustering⁹² with a batch size of 8000 to define 10 prototypical neighborhoods from the 53 million extracted histograms. Each cell was then assigned to the closest prototype using nearest-neighbour classification. We subsequently aggregated the data for each tissue spot to quantify how many cells corresponded to each identified niche. ”

b. Aggregation of niches for survival analysis uses a MIL framework. What is the input and output of the MIL ?

We thank the reviewer for this important question and apologize for the lack of clarity in the original description of the multiple instance learning (MIL) framework used for survival analysis.

In our approach, MIL takes as input all available tumor cores from a given patient, each represented by its niche composition features. Using a maximum-over-instances rule, the model outputs a single aggregated risk score per patient. This reflects a "hot-spot approach", assuming that the core with the highest risk score carries the most prognostically relevant information. We have clarified this point in the revised Methods section.

Methods section (p. 36):

[revised] *“Patients are represented by up to four tissue spots. We analyzed them using a multiple instance learning approach and chose a late fusion strategy for better interpretability. In this approach, each spot was evaluated individually for survival based on its niche composition features, and the model outputs a continuous risk score (logit) for each spot. To derive a patient-level risk estimate, we applied max-pooling across all spot-level outputs – meaning the highest risk score among a patient’s spots determines the overall risk classification (“hot-spot approach”). This reflects the clinical rationale that the most aggressive region within the tumor is most prognostically relevant. The final patient-level prediction is thus determined by the maximum risk score across all available spots.”*

c. It is not clear how RS1, RS2, and RS3 are labeled for training of the Cox regression model for survival prediction.

We thank the reviewer for this clarification request. RS1, RS2, and RS3 were not used as predefined labels during model training. The Cox regression model was trained on continuous survival data from the Berlin cohort without stratification. The patient risk strata (RS1–RS3) mentioned in the manuscript were derived post hoc by dividing the predicted risk scores into tertiles. This discretization was introduced solely to improve interpretability in visualizations and subgroup analyses. The use of three risk groups offers a natural counterpart to the UICC clinical staging system for earlier stage cancers (stages I–III), according to which patients in our study were classified. We have clarified this procedure in the revised Methods section to improve transparency.

Methods section (p. 36):

[revised] *“We collected the risk attributions generated by the Cox regression model and stratified the test cohort into three groups based on the distribution of these scores. Specifically, we divided the patients into tertiles corresponding to the lower, middle, and upper thirds of predicted risk. These three groups are referred to in the main text as risk strata RS1 (low risk), RS2 (intermediate risk), and RS3 (high risk). This post hoc discretization was introduced to enhance the interpretability of the survival analysis and visualization, while maintaining the model’s continuous output during training and inference.”*

4. Cell classification: if the algorithm learns features from H&E based on IF annotations, why does it need the IF during inference ? What features does the algorithm learn ?

We thank the reviewer for raising this question regarding the complementary roles of mIF and H&E in the cell classification process. Both, H&E and mIF, are needed during inference and are used by an H&E-based segmentation model and IF-based cell type models respectively. The following describes this in more detail.

The segmentation model learns to extract morphological features from H&E images that are informative for identifying tissue regions (e.g., carcinoma, stroma, or non-neoplastic tissue) and for distinguishing certain cell populations. Specifically, it can differentiate carcinoma cells from non-neoplastic epithelial cells or from cells that may express cytokeratins artifactually – such as sometimes macrophages – based on morphological features like nuclear shape, cytoplasmic texture, and spatial organization. This is particularly relevant because both malignant and non-malignant epithelial-like cells express cytokeratins and would otherwise be difficult to distinguish using mIF alone. This model was trained with pathologist annotations based on H&E.

At the same time, there are cell types that cannot be reliably classified based on H&E morphology alone. A prominent example is lymphocytes: while they are clearly identifiable in H&E, differentiating between B cells and T cells requires specific marker information provided by mIF (e.g., CD20 vs. CD3). To further differentiate the cell types, one model per IF channel was trained to predict antigen positivity. These models were trained with pathologist annotation based on the respective IF marker.

Thus, both modalities contribute complementary information – H&E provides morphological context and carcinoma detection, while mIF enables precise phenotypic identification based on marker expression. We have clarified this point in the revised Methods section.

Methods section (p. 33):

[revised] *“The integration is required for two reasons. 1) The H&E-based model enables tissue-level segmentation into carcinoma, stroma, necrosis, and non-neoplastic regions, thereby ensuring that the analysis focuses on histologically relevant compartments. This kind of spatial annotation cannot be reliably obtained from mIF alone due to its limited morphological information. 2) The mIF-based models, in turn, contribute precise cell-type classification through*

antigen-specific labeling, which is essential for phenotypic resolution. The combined use of H&E and mIF is particularly critical for distinguishing morphologically similar cell types. For example, carcinoma cells and normal epithelial cells such as pneumocytes or ciliated epithelial cells often share expression of epithelial markers (e.g., cytokeratins), making them difficult to differentiate using mIF alone. Here, the morphological features captured by H&E – such as nuclear atypia, cell shape, and architectural arrangement – are used for accurate classification. Conversely, certain immune cell subtypes (e.g., B vs. T lymphocytes) are indistinguishable based on H&E but can be clearly resolved using mIF markers such as CD20 and CD3. Thus, the integration of both modalities provides complementary strengths: H&E delivers detailed tissue architecture and tumor morphology, while mIF enables high-resolution immunophenotyping. Together, they allow robust and context-aware cell classification across diverse tissue environments.”

5. Questions / comments related to figures:

a. Figure 1. I) Does the composition of the immune cell infiltrate change with tumor cell content ? II) What are the purple cell populations in panel A ? III) Please use the same color for subpopulations of immune cells in panel A and B.

We thank the reviewer for these points and assume they refer to Figure 2 rather than Figure 1.

I) Relationship between immune cell composition and tumor cell content:

Our data show that immune cell composition varies both in relation to and independently of tumor cell density. For instance, LUAD tumors in Cluster Groups 3 and 6 (Figure 2E, Supplementary Table S5) display similar tumor cell densities (1115/mm² vs. 1155/mm²), but differ markedly in their macrophage-to-lymphocyte ratios (0.32 vs. 0.89). Cluster Group 5, with a much higher tumor cell content (2340/mm²), shows a further increase in this ratio (1.85). This indicates that immune composition is not solely a function of tumor cell content, but also reflects tumor-intrinsic features and their interaction with the microenvironment.

II) Purple populations in Figure 2A:

These cells correspond to immunosuppressive carcinoma cells (iCC). In the updated plots, this population is displayed in bright turquoise as part of the adjusted color scheme (see point III).

III) Color scheme in panels A and B:

We thank the reviewer for this helpful suggestion. In the revised Figure 2 (Rebuttal Figure 3), we harmonized the color scheme so that subpopulations of immune cells are shown consistently across panels A and B.

b. Figure 1: C. poor H&E quality and green artifacts. Please improve quality of images.

We thank the reviewer for pointing this out and apologize for the suboptimal image quality in the original submission.

We have carefully reviewed and improved the quality of all H&E images throughout the manuscript, including the removal of artifacts and the optimization of contrast and resolution.

This applies in particular to Figure 1c, which has been revised accordingly. We hope that the improved image quality now meets the expected standards and enhances visual clarity.

c. Figure 1: E and G: these panels are not informative because many cell types show no difference between clusters. Show cell phenotypes that drive the clustering and that separate the clusters.

We thank the reviewer for this helpful observation. We assume the comment refers to **Figure 2e and 2g**, where the heatmaps show the distribution of all identified cell phenotypes across the defined clusters. Our intention in these panels was to provide a comprehensive overview of the full landscape of detected cell phenotypes in LUAD and LUSC. However, we agree that, due to the wide variation in phenotype abundance, visual differences between clusters may appear subtle or less informative for certain cell types. To improve clarity and reader guidance, we have therefore added visual annotations to highlight key patterns in the heatmaps and enlarged the cell type labels to enhance legibility (see revised Figure 2; **Rebuttal Figure 3**).

We hope these changes substantially improve interpretability. If the reviewer still finds the current format insufficient, we would be happy to consider moving the full heatmaps to the Supplement and replacing them with focused heatmaps showing only the most abundant and cluster-defining phenotypes. Please note, however, that this would entail changes in the clustering results due to feature reduction.

Rebuttal Figure 3 (also revised Figure 2): Cell phenomics overview

- (A) Stacked bar plots of the abundances of cell phenotypes* for each tumor sample in LUAD showing the highly variable distribution of immune cells and cancer cells across tumors.
- (B) Same as (A) but for LUSC. In contrast to LUAD, a noticeable higher number of tumors exhibited high densities of immunosuppressive carcinoma cells (iCC).
- (C) Examples of tumors with high and low infiltration of immune cells and relatively low and high carcinoma cell content in LUAD.
- (D) Same as (C) but for LUSC.
- (E) Heatmaps sorted with hierarchical clustering in LUAD (z-scored and log-transformed for improved visibility). Horizontal axis: tumor samples; vertical axis: cell types. Arabic numbers highlight clusters corresponding to different clinically relevant immune states (cf. F). Visual annotations highlight key patterns within the immune states.
- (F) Tumor examples with different clinically relevant immune states in LUAD.
- (G) Heatmaps sorted with hierarchical clustering in LUSC (z-scored and log-transformed for improved visibility). Horizontal axis: tumor samples; vertical axis: cell types. Arabic numbers highlight clusters corresponding to different clinically relevant immune states (cf. H). Visual annotations highlight key patterns within the immune states.
- (H) Tumor examples with different clinically relevant immune states in LUSC.

*Figure shows cell phenotype name abbreviations. For full names see Supplementary Table S2.

Minor concerns:

The classification of carcinomas by immune status refers to references between 2011 and 2026 (Schalper et al. 2015; Mahmoud et al. 2011 and Taube et al. 2012).

We thank the reviewer for pointing this out. We have corrected the citation accordingly and now refer specifically to Mahmoud et al. (2011) and Taube et al. (2012), who originally introduced and established the concept of immune-based classification of carcinomas in relation to prognosis and treatment response. This has been updated in the revised manuscript.

Results section (p. 14):

[revised] *“The classification of carcinomas by their specific immune status for prognosis and treatment response has been established by Mahmoud et al. 2011³⁵ and Taube et al. 2012³⁶.”*

- Is the methodology sound? Does the work meet the expected standards in your field?

The methodology is sound.

- Is there enough detail provided in the methods for the work to be reproduced? Please see major concerns for questions that need to be answered to reproduce this work.

Reviewer #4

The repository appears to be well structured. The README file is comprehensive. The code is executable according to the provided instructions, and results obtained. However, the results are based on a subset of the full data set used for the paper. The full data is described as "available upon request", however it is not sure if all requests will be granted, or there will be additional criteria for granting access.

We thank the reviewer for carefully assessing the repository and for confirming that the provided code and subset data are well structured and reproducible.

Regarding the availability of the full dataset: due to storage constraints (>1 TB), it was not feasible to host all raw image files directly on Zenodo. To balance accessibility with practical limits, we have made all processed cell-classification results, niche annotations, and 14,000 high-resolution image tiles openly available on Zenodo. These resources are sufficient to reproduce the analyses presented in the manuscript.

For researchers requiring access to the complete raw dataset, we have established a transparent and straightforward request process. Access is provided via a secure S3 bucket, with the sole requirements being (i) a signed non-commercial use declaration and (ii) an academic email address. No scientific or institutional gatekeeping is applied, and all bona fide academic requests will be granted. This ensures open scientific use while allowing responsible handling of large-scale sensitive data.

We have clarified this procedure in the Data and software availability section (p. 39) of the manuscript to avoid ambiguity.

[added] *"All processed cell-classification results and niche annotations are publicly available on Zenodo (DOI: 10.5281/zenodo.11395885). High-resolution image tiles for each patient, including all multiplex immunofluorescence (mIF) channels and H&E (14,000 images), are provided in a second Zenodo repository (DOI: 10.5281/zenodo.16882468). The complete raw dataset (>1 TB), including patient metadata and full image files, is available via a secure S3 bucket upon request to the corresponding author. Access requires only a signed non-commercial use declaration and an academic email address; all bona fide academic requests will be granted. Code is available at https://github.com/gabrieldernbach/cell_niches."*

Reviewer #5
